# Energy-efficient high-fidelity image reconstruction with memristor arrays for medical diagnosis

Han Zhao [1,3], Zhengwu Liu [1,3], Jianshi Tang [1,2] ✉, Bin Gao [1,2], Qi Qin[1], Jiaming Li[1], Ying Zhou[1], Peng Yao[1], Yue Xi[1], Yudeng Lin[1], He Qian[1,2] & Huaqiang Wu [1,2]

Medical imaging is an important tool for accurate medical diagnosis, while state-of-the-art image reconstruction algorithms raise critical challenges in massive data processing for high-speed and high-quality imaging. Here, we present a memristive image reconstructor (MIR) to greatly accelerate image reconstruction with discrete Fourier transformation (DFT) by computing-in-memory (CIM) with memristor arrays. A high-accuracy quasi-analogue mapping (QAM) method and generic complex matrix transfer (CMT) scheme was proposed to improve the mapping precision and transfer efficiency, respectively. High-fidelity magnetic resonance imaging (MRI) and computed tomography (CT) image reconstructions were demonstrated, achieving software-equivalent qualities and DICE scores after segmentation with nnU-Net algorithm. Remarkably, our MIR exhibited 153× and 79× improvements in energy efficiency and normalized image reconstruction speed, respectively, compared to graphics processing unit (GPU). This work demonstrates MIR as a promising high-fidelity image reconstruction platform for future medical diagnosis, and also largely extends the application of memristor-based CIM beyond artificial neural networks.

Medical imaging has been widely used to diagnose or monitor patients' medical conditions by revealing the internal structures of organs, such as lung and brain, to identify the lesions and perform surgical interventions[1–4]. Magnetic resonance imaging (MRI) and computed tomography (CT) are two representative medical imaging technologies[5–8]. Although adopting different imaging principles, their imaging processes can be roughly divided into two fundamental steps: signal acquisition and image reconstruction (Fig. 1b, c). In the signal acquisition stages, MRI uses radio frequency coils to collect signals from the protons in a body under the magnetic fields[5,9,10], while CT uses X-ray detectors to receive signals from a beam of rays that pass through the patient's body[7,11,12]. In the image reconstruction stage, both MRI and CT use specific algorithms to reconstruct medical images for

regions of interest, where many algorithms are based on Fourier transformations since the acquired data are usually represented in the Fourier space[13–15]. To meet the demand for better image quality and higher imaging speed, the number of MRI radio frequency coils and CT detectors has been increased dramatically[4,16–20], resulting in an explosive growth of raw data to be processed. Besides, more sophisticated reconstruction algorithms based on iteration, deep learning and data-adaptive methods are also being implemented[21–23]. Amid the slowdown of Moore's law scaling[24,25], such computationally intensive tasks impose critical challenges for conventional computing hardware[5,26] based on von Neumann architecture with physically separated computing and memory units, limiting their energy efficiency. Thus, the speed and energy consumption of the image reconstruction step has

[1]School of Integrated Circuits, Beijing National Research Center for Information Science and Technology (BNRist), Tsinghua University, Beijing 100084, China. [2]Beijing Innovation Center for Future Chips (ICFC), Tsinghua University, Beijing 100084, China. [3]These authors contributed equally: Han Zhao, Zhengwu Liu. ✉ e-mail: jtang@tsinghua.edu.cn

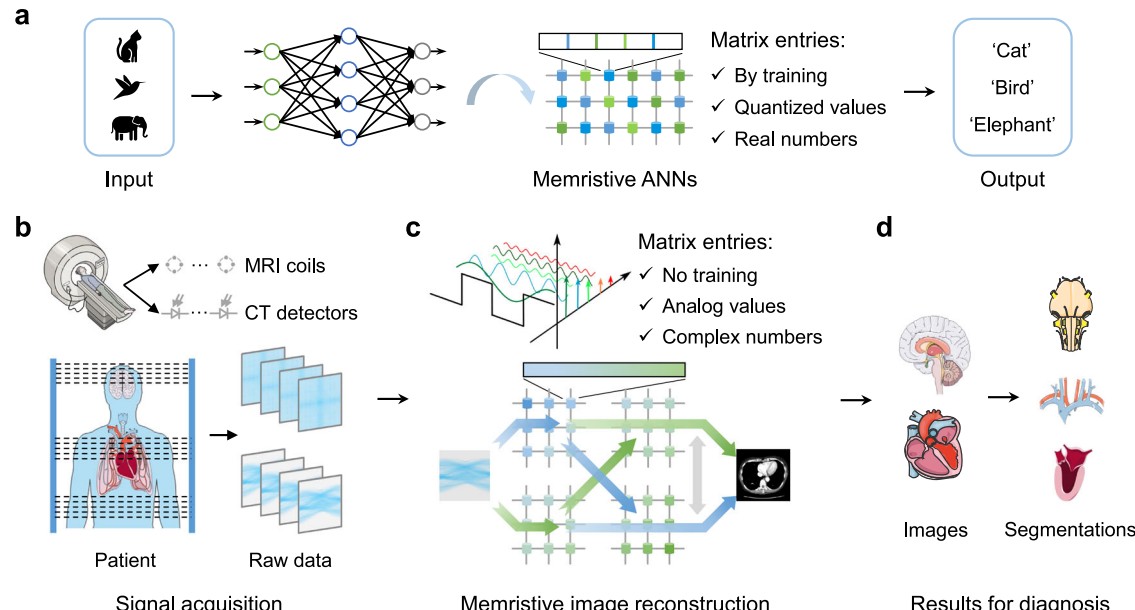

**Fig. 1 | Illustration of memristive artificial neural networks (ANNs) and medical imaging system. a** Memristive ANNs. The matrix entries of ANNs are obtained by training and they are usually quantized before being mapped to memristor arrays, resulting in quantization errors. Besides, most ANNs are computed in real number fashion. **b–d** Memristive medical imaging system. **b** Medical signal acquisition. Explosive amount of raw data is acquired from medical scanners such as magnetic resonance imaging (MRI) and computed tomography (CT). **c** Memristive image reconstruction. The matrix entries used in signal processing algorithms here are pre-calculated without training, making them more susceptible to the non-ideal device characteristics of memristors. In addition, their entries are usually expressed in analogue manner with both real and imaginary parts, requiring a completing different mapping strategy onto memristor arrays. **d** Results for medical diagnosis. Medical images of human body are reconstructed from raw data and then further segmentation and diagnosis can be performed. The cartoon pictures of human organs and medical equipment used in **b** and **d** was partly generated using Servier Medical Art, provided by Servier, licensed under a Creative Commons Attribution 3.0 unported license.

become a serious bottleneck for the development of portable medical imaging systems.

Fortunately, computing-in-memory (CIM) technology based on emerging nonvolatile memories, such as memristors, can provide an alternative solution for medical image reconstruction with ultrahigh efficiency to break the von Neumann bottleneck. In this paradigm, computations are carried out at the place where data are stored through physical laws, largely reducing the energy-intensive data movement[27–29]. In recent years, memristive CIM paradigm has been widely used in implementing artificial neural networks (ANNs) (Fig. 1a), showing appealing advantages in terms of energy efficiency and speed compared to conventional hardware[30–32]. Besides ANNs, there have also been attempts to use memristor arrays for implementing classic signal processing algorithms[33], such as finite impulse response (FIR) filter[34] and discrete Fourier transformation (DFT)[35,36], which has the potential to significantly accelerate medical image reconstruction speed and reduce energy consumption. In both applications, the most computationally intensive computations are vector-matrix multiplication (VMM); however, their actual implementations on memristor arrays are quite different in two aspects. Firstly, the entries in the matrix, i.e., synaptic weights, for ANNs are trainable, so that the effect of non-ideal device characteristics of memristors (e.g., device noise and conductance fluctuation) may be accommodated by training[30,31]. In comparison, the entries in the matrix for signal processing algorithms, e.g., DFT coefficients, are pre-calculated without any training, so they are more susceptible to those non-ideal device characteristics, which could lead to large errors in the output. Secondly, while the entries in the matrix for ANNs are usually real numbers, the matrices used in signal processing usually have both real and imaginary parts, and directly mapping them onto different memristor arrays (as proposed in literature[35,36]) could result in large overhead in both energy consumption and area.

Therefore, a new scheme to efficiently implement signal processing algorithms such as DFT on memristor arrays needs to be developed.

In this work, we propose and demonstrate a memristive image reconstructor (MIR), whose core is a memristive DFT (Fig. 1c). To efficiently implement DFT on memristor arrays, we have developed two strategies, quasi-analog mapping (QAM) and complex matrix transferring (CMT) scheme, to improve the mapping precision and transfer efficiency, respectively. With QAM, memristive DFT results achieves better consistency with software-calculated results, compared with the conventional quantized mapping (QM) method. With CMT, memristive DFT also has a smaller overhead of peripheral circuits, compared to conventional real-imaginary separated scheme. Based on the above two strategies, MIR is used to experimentally demonstrate MRI and CT image reconstruction tasks, achieving software-equivalent image reconstruction peak signal-to-noise ratios (PSNR) of 40.21 dB and 22.38 dB for MRI and CT images, respectively. To validate that the MIR-reconstructed images meet the requirements of medical diagnosis, we utilize the widely recognized biomedical image segmentation algorithm, nnU-Net[37,38], to segment and extract organs from MIR and software-reconstructed images, achieving similar DICE scores around 0.98. Besides, MIR shows 153× advantage in energy efficiency and 79× advantage in normalized image reconstruction speed for CT reconstruction task, compared to CMOS systems, indicating its great potential in low-power and high-speed portable medical imaging system for future medical scenarios.

## Results

### Implementation of MIR on memristor array
We construct the MIR based on our customized memristor hardware platform (Fig. 2a)[31], which consists of eight 2K-cell memristor arrays. Figure 2b shows the transmission electron microscope (TEM) image of the fabricated memristor array with unit cells of one-transistor-

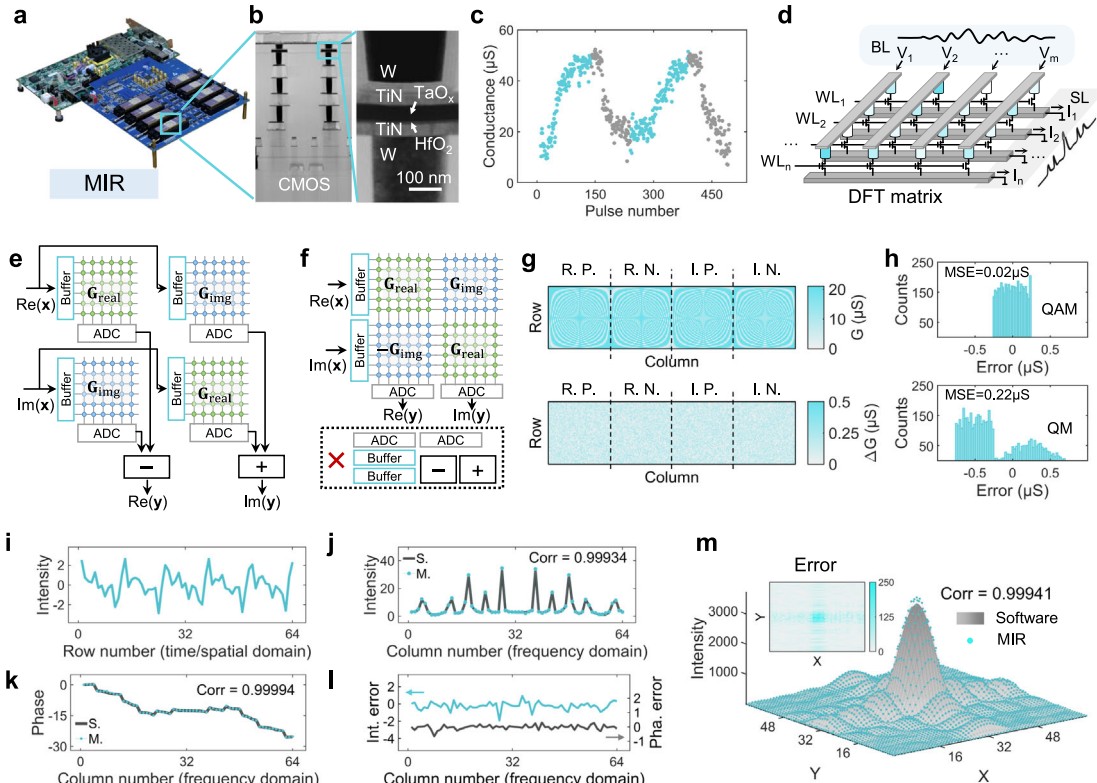

**Fig. 2 | The implementation of memristive image reconstructor (MIR).**
**a** Photograph of the MIR system with eight 2 K memristor chips and an FPGA board.
**b** TEM images of the memristor array (left) and device (right). **c** Analog switching characteristics of a typical memristor device, where the SET voltage is 1.4 V and the RESET voltage is 1.5 V. **d** Implementation of discrete Fourier transform (DFT) on memristor array. DFT matrix is initially mapped onto a memristor array as the conductance. Then time-domain signals are fed into the bit lines (BLs) of memristor array as voltage pulses and frequency-domain signals are calculated as the output currents from the source lines (SLs). The word lines (WLs) are connected to the gate of transistors to select memristor cells. **e** Conventional DFT implementation on memristor arrays with extra copies of peripheral circuits to implement the addition and subtraction operations as well as analog-to-digital conversion (ADC). Re, the real part. Im, the imaginary part. **f** Proposed DFT implementation with complex matrix transfer (CMT) scheme on memristor array, saving more than half of

peripheral circuit overhead. **g** Top, experimental mapping results with quasi-analog mapping (QAM) strategy. Bottom, the difference between the target conductance and experimentally mapped values. R.P. and R.N. denote the positive and negative matrices of the real part of the DFT matrix, respectively. I.P. and I.N. denote the positive and negative matrices of the imaginary part, respectively. **h** Mapping error distribution of QAM and quantized mapping (QM). MSE, mean squared error. **i–l** Comparison between the DFT results obtained by software (S.) and memristor (M.). Time-domain or spatial-domain signals (**i**) are transformed into frequency-domain signals, which consist of two parts, the intensity (**j**) and the phase (**k**). The difference between the intensity and phase of DFT results obtained by software and memristor are shown in **l**. Corr, correlation coefficient. **m** Comparison between 2D frequency-domain results computed by software and memristive 2D DFT. Error distribution is given in the inset.

one-resistor (1T1R). The memristor device has a material stack of TiN/TaOₓ/HfO₂/TiN, where HfO₂ is the resistive switching layer and TaOₓ serves as the thermal enhanced layer to improve the analog switching characteristics (Fig. 2c)[39,40]. The conductance of memristor can be continuously modulated by SET or RESET voltage pulses, allowing us to map the matrix entries onto memristor conductance precisely.

To demonstrate the functionality of the MIR system, we first implement a 64-point one-dimensional (1D) complex DFT on the memristor arrays with 16 K memristors. DFT is a widely used time/spatial-frequency analysis technology and plays a key role in many applications including denoising, imaging and communication. As shown in Fig. 2d, the computation of DFT can be regarded as a multiplication of time/spatial-domain input signal and a DFT matrix, and the output result following the Ohm's law and Kirchhoff's current law represents the frequency-domain signal. In this way, the time complexity of DFT computation can be significantly reduced down to $O(1)$, representing a dramatic speed-up compared to $O(N^2)$ complexity for conventional DFT implementation or $O(N\log N)$ complexity for fast Fourier transformation (FFT).

As the DFT coefficients matrix has both real and imaginary parts, conventionally, four independent memristor arrays of the same size $N \times N$ would be needed to map the matrix entries onto memristor

conductance for the implementation of a $N$-point DFT[35,36], as shown in Fig. 2e. Besides memristor arrays, such implementation would also need extra copies of peripheral circuits to implement the addition and subtraction operations as well as analog-to-digital conversion (ADC), resulting in additional hardware overhead and computing latency. To circumvent this problem, here we propose a generic complex matrix transfer (CMT) scheme. As shown in Fig. 2f, by assembling the four same-size DFT matrices into an integrated matrix, both the real and imaginary parts of DFT results can be directly obtained in a single step because the addition and subtraction operations can be realized inside the memristor array rather than by peripheral circuits, reducing the computing latency. In the meanwhile, the number of peripheral circuits such as ADCs and buffers can also be reduced by at least one half, saving both energy and area cost. In addition, inverse DFT (IDFT) can also be implemented using such CMT scheme in a similar way as DFT. The mathematical derivation of DFT and IDFT implementations on memristor arrays are described in the Methods section.

In addition, to improve the computing accuracy of the memristive DFT, it is necessary to improve the mapping precision of DFT matrix entries, whose values are pre-calculated rather than obtained by training and thus are more susceptible to the mapping errors than

ANNs. Different from the commonly used QM strategy for mapping synaptic weights of ANNs, here we develop a more accurate quasi-analog mapping (QAM) strategy to eliminate the quantization error[41] (more information can be found in the Methods section and Supplementary Fig. 1). Figure 2g shows the mapping results (top panel) and mapping error (bottom panel) with QAM, indicating an excellent mapping precision. Here, the real part (R) and imaginary part (I) of DFT matrix are mapped onto the memristor array in the form of differential pairs (P, positive part; N, negative part). To make a comparison, Fig. 2h plots the distribution of mapping error for a 64-point DFT matrix with QAM and QM (see Supplementary Fig. 1 for the breakdown of mapping error). The mean squared error (MSE) of QAM is 0.02 μS, which is much lower than that of QM (0.22 μS), suggesting significantly enhanced mapping precision of QAM. In fact, according to our experimental results shown in Supplementary Fig. 4, the implementation of a greater number of DFT point on MIR shows even higher consistency with software-computed results, indicating a better signal processing quality. However, as the number of DFT point grows, the required number of memristor conductance levels also increases rapidly (for example, 64-point DFT requires 25 conductance levels), and thus the efficacy of our quantization-error-free QAM strategy can be more notable to improve the mapping precision and computing accuracy.

Figure 2i-l give a typical example of memristive DFT and compare the results with software-based DFT. The time-domain signal (Fig. 2i) is fed into MIR and transformed into frequency-domain signal, which consists of two parts, intensity (Fig. 2j) and phase (Fig. 2k). The difference between DFT results from memristor and software is shown in Fig. 2l. It can be seen that these two results match well and the correlation coefficients of frequency-domain intensity and phase are as high as 0.99934 and 0.99994, respectively. Furthermore, using the MIR, a more sophisticated two-dimensional DFT (2D DFT) is demonstrated. Taking a $64 \times 64$ image for example, up to 128 64-point DFTs are required to obtain the 2D DFT results. During the entire 2D DFT computing process, the error of each DFT affects the subsequent computation and hence there is an even higher demand of computing accuracy for 2D DFT. Figure 2m shows the comparison of 2D frequency-domain results computed by software and memristive 2D DFT, exhibiting a high correlation coefficient of 0.99941. This result suggests excellent computing accuracy of memristive 2D DFT, thanks to the good analog switching characteristics and QAM strategy. Similarly, IDFT and 2D IDFT can also be implemented on MIR (see Methods).

## MRI image reconstruction with MIR
To evaluate the overall performance of our MIR in data-intensive medical imaging applications, we further implement an end-to-end MRI image reconstruction and segmentation task. Figure 3a illustrates the complete MRI data processing procedure. Raw data from MRI scanner are sampled in K-space (2D Fourier space) and then fed into MIR. 2D IDFT is then performed to transform the data into human body slice images in spatial domain (reconstructed MRI images). Next, medical image segmentation is usually performed to annotate the regions of interests for medical diagnosis, where the segmentation quality largely affects the diagnosis accuracy. Here, in this work, the non-ideal device characteristics of memristors and arrays, such as read noise (as shown in Supplementary Fig. 2), stuck-at faults (as shown in Supplementary Fig. 6), mapping error and interconnect resistance, could degrade the quality of reconstructed images by MIR. To verify if the key information in MIR images is preserved, we use nnU-Net to process the reconstructed images from both software and MIR and obtain segmented human organs or tissues, whose results are used for comparison to examine the quality of reconstructed images and the computing accuracy of MIR.

Here we use the heart dataset provided by King's College London[42], which contains 30 three-dimensional (3D) MRI images

covering the entire human heart. As the ground truth, the left atrium was segmented by an expert using an automated tool followed by manual corrections. Figure 3b shows the raw data from MRI scanner sampled in K-space. Through our memristive 2D IDFT, a series of sagittal plane images are reconstructed (Fig. 3c) to finally obtain a 3D MRI image. Figure 3d, e shows the MRI images observed from different angles (transverse plane image and coronal plane image), indicating good consistency of different sagittal planes and high-quality image reconstruction by MIR. The reconstructed MRI images and segmented results (left atrium) are shown in Fig. 3f (software) and Fig. 3g (MIR), showing good consistency and visually no obvious difference between the two segmentation results. Quantitatively, an average PSNR of 40.21 dB and signal-to-noise ratio (SNR) of 24.14 dB is achieved for MIR as shown in Fig. 3h. Similar values are achieved on 20 other MRI datasets (Fig. 3i), validating the high-fidelity image reconstruction by MIR. As a reference, a PSNR value over 30 dB indicates that it is hard to tell the difference between the original and reconstructed images by human naked eyes. Furthermore, the MIR and software reconstructed images receive a similarly high DICE score -0.98 after segmentation with nnU-Net as shown in Fig. 3j. These results suggest that MIR exhibits excellent performance in the MRI image reconstruction task even in the presence of memristor read noise and mapping error. The benchmark in Fig. 3k-l further reveals that MIR is 112× higher in energy efficiency and 36× higher in normalized image reconstruction speed than Nvidia Tesla V100 GPU, respectively (see Methods section for more details).

## CT image reconstruction with MIR
In the above demonstration of MRI image reconstruction, only one step of 2D DFT is carried out. In more advanced medical imaging algorithms, more than one steps of transformation are needed, where the impact of cumulative errors could degrade the image quality. As shown in Supplementary Fig. 3, with more steps of DFT on MIR, the signal distortion tends to increases. To further examine the computing accuracy and noise robustness of MIR, here we implement a more complicated task, CT image reconstruction based on Fourier central slice theorem, which contains 3 steps of DFT/IDFT (1 step for 1D DFT and 2 steps for 2D IDFT). The data processing procedure is shown in Fig. 4a. During CT scan, X-ray projections of human body from various angles are processed to produce cross-sectional images (slices) for diagnostic and therapeutic purposes. Each projection vector is fed to MIR for DFT, yielding frequency-domain signal in 2D Fourier space. Then reconstructed CT images can be obtained by performing 2D IDFT on the 2D Fourier space. During the entire CT image reconstruction process, three steps of DFT in total are required to transform the raw projections into CT images. Finally, organs or tissues can be segmented by nnU-Net from the reconstructed CT images. As an example, Fig. 4b shows the original human organ and reconstructed CT images[43] along with intermediate results.

Here, we use the spleen dataset from Memorial Sloan Kettering Cancer Center, including 61 portal venous phase CT scans[44]. In this dataset, the spleen was segmented semi-automatically and then were manually modified by an expert abdominal radiologist, serving as the ground truth. Figure 4c, d show the reconstructed CT images by software and MIR, and the segmented parts (spleen) from nnU-Net are also annotated. Visually, MIR reconstructed CT images preserve most critical information even in the presence of the above-mentioned cumulative errors. In Fig. 4e, we put the segmented 3D spleen models of software and MIR together and examine them from 4 different angles. Only pixel-level difference can be observed on the spleen surface, indicating excellent consistency. Quantitatively, in Fig. 4f, g, the PSNR of CT images reconstructed by software and MIR are 22.52 dB and 22.38 dB, respectively, and the DICE score of segmented results are 0.985 and 0.977 for software and MIR,

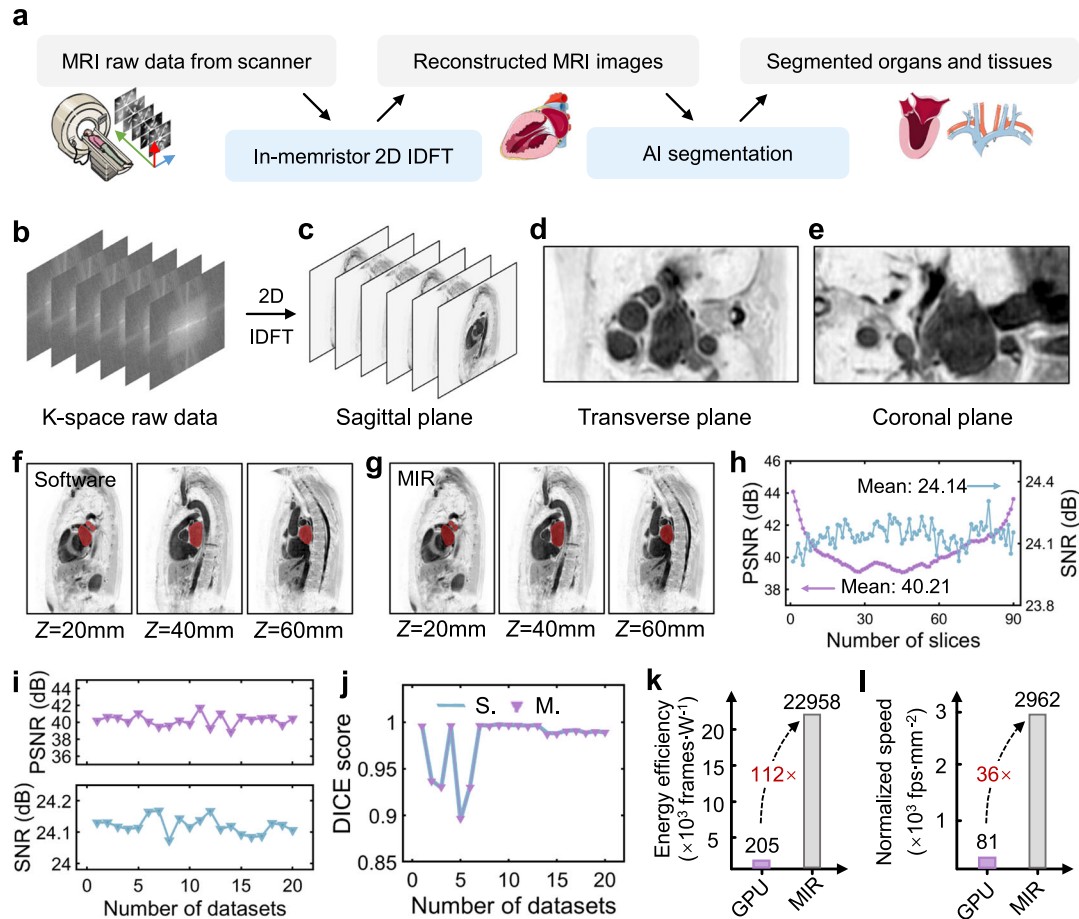

**Fig. 3 | MRI image reconstruction with MIR. a** Schematic of MRI image reconstruction with MIR followed by AI segmentation. **b** K-space raw data from MRI scanner. **c** Reconstructed sagittal plane images, computed with 2D IDFT by MIR. Combining all the sagittal plane images together, a 3D MRI image can be obtained. **d**, **e** Transverse plane image (**d**) and coronal plane image (**e**) of the reconstructed 3D MRI image. **f**, **g** Segmented left atriums from nnU-Net, with software-reconstructed images (**f**) and MIR-reconstructed images (**g**). Z, the coordinate on the z-axis which goes from inferior to superior. **h** The PSNR and SNR of each reconstructed MRI slice of No.16 dataset by MIR. **i** The reconstructed image quality for 20 MRI datasets. **j** DICE scores of nnU-Net segmentation results of software-reconstructed (S.) and MIR-reconstructed (M.) images for 20 MRI datasets. **k** The comparison of energy efficiency of GPU and MIR. **l** The comparison of the normalized image reconstruction speed of GPU and MIR. The cartoon pictures of human organs and medical equipment used in **a** was partly generated using Servier Medical Art, provided by Servier, licensed under a Creative Commons Attribution 3.0 unported license.

respectively. These results reaffirm that the key information contained in CT raw projections is well preserved and our MIR shows superior robustness to cumulative errors and memristor device noise. Furthermore, the benchmark in Fig. 4h, i show that MIR again is 153× higher in energy efficiency and 79× higher in normalized image reconstruction speed than GPU, respectively (see Methods section for more details).

## Discussion

In conclusion, we have proposed and experimentally demonstrated a memristive medical image reconstructor, MIR, which shows excellent performance in energy efficiency and computing speed and is also highly robust to the non-ideal device characteristics of memristors. With the developed QAM strategy and CMT method, MIR could efficiently implement high-accuracy 1D/2D DFT and IDFT. In the MRI and CT image reconstruction tasks, compared with Nvidia Tesla V100 GPU, MIR has achieved a software-equivalent image reconstruction quality in terms of PSNR and DICE score, and meanwhile exhibited 112× and 153× improvements in energy efficiency, and 36× and 79× improvements in the normalized image reconstruction speed, respectively. Such computational advantages could be further improved by using larger memristor arrays and further optimizing the memristor analog

switching characteristics. This work suggests that our MIR system could provide a high-speed and energy-efficient computing platform for future medical imaging technology, which paves the road for low-power portable and point-of-care diagnostics. It also helps widely extend the application of memristor-based CIM beyond ANNs.

## Methods

### Fabrication and programming of 1T1R memristor array

The memristor array was fabricated with a standard 0.13μm Si CMOS process. Each 1T1R cell consists of a memristor and a Si transistor. The memristor device has a material stack of $TiN/TaO_x/HfO_2/TiN$. 30nm-thick TiN was sputtered as the top and bottom electrodes. 8nm-thick $HfO_2$ was deposited by atomic layer deposition (ALD) as the resistive switching layer. 45 nm-thick $TaO_x$ was sputtered as the thermal enhanced layer to improve the analog switching characteristics[40,45]. As for the programming of 1T1R memristor array, we use the standard write-verify programming scheme[45]. Here, multiple voltage pulses are applied to the 1T1R memristor cell to increase (decrease) the conductance, until the conductance is larger (smaller) or equal to the target values. This process is repeated until the memristor conductance to programmed within the error margin of the target value.

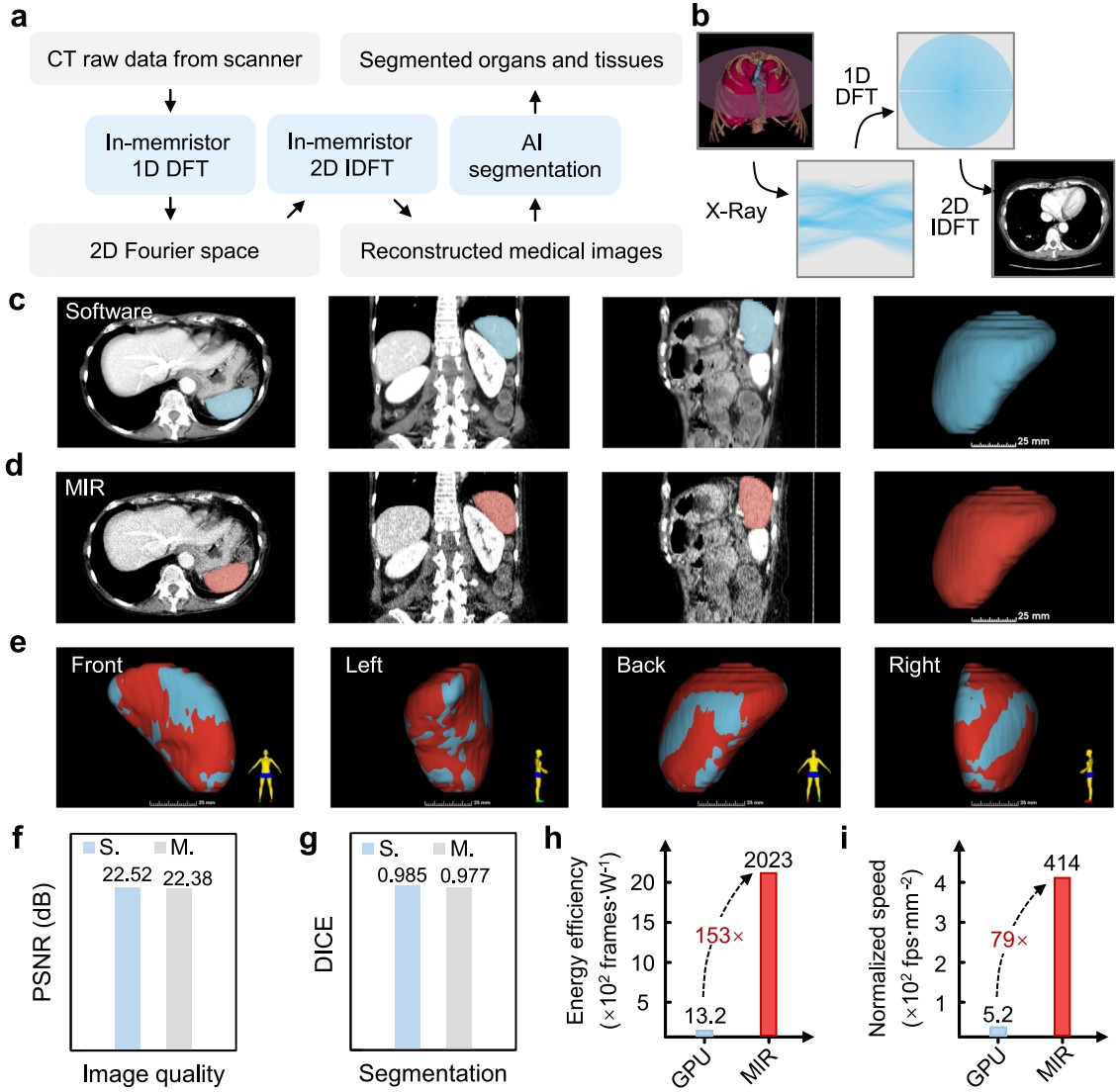

**Fig. 4 | CT image reconstruction with MIR. a** Schematic of CT image reconstruction with MIR followed by AI segmentation. **b** Illustration for the CT image reconstruction task. Sub-figures, from left to right, are actual human organ (the pink disc represents the section where the CT slice is acquired), projections from CT X-ray scanner, 2D Fourier space signal (intermediate results) and reconstructed CT image. **c**, **d** Software-reconstructed and MIR-reconstructed 3D CT image and its segmentation results from nnU-Net (the spleen is labeled in blue and red). Sub-figures, from left to right, are sagittal plane image, transverse plane image, coronal plane image and 3D model (segmented spleen only). **e** Comparison between the segmented spleens with software-reconstructed (blue) and MIR-reconstructed (red) 3D CT images. **f** Comparison of image quality reconstructed by software (S.) and MIR (M.). **g** DICE score of nnU-Net segmented results from software-reconstructed and MIR-reconstructed images. **h** The comparison of energy efficiency of GPU and MIR. **i** The comparison of the normalized image reconstruction speed of GPU and MIR.

## Implementation of 1D DFT on CMT scheme

Mathematically, DFT converts a complex $N$-point sequence $\mathbf{x}_n$ into a complex sequence $\mathbf{X}_k$ of the same length:

$$\mathbf{X}_k = \frac{1}{\sqrt{N}}\mathbf{W}\cdot\mathbf{x}_n \qquad (1)$$

Here, $\mathbf{X}_k \in \mathrm{R}^{N\times 1}$, $\mathbf{x}_n \in \mathrm{R}^{N\times 1}$ and $\mathbf{W} \in \mathrm{R}^{N\times N}$. $\mathbf{W}$ is the DFT matrix which is defined by $\omega = e^{-2\pi i/N}$ as:

$$\mathbf{W} = \frac{1}{\sqrt{N}}\begin{bmatrix} 1 & 1 & 1 & 1 & \cdots & 1 \\ 1 & \omega & \omega^2 & \omega^3 & \cdots & \omega^{N-1} \\ 1 & \omega^2 & \omega^4 & \omega^6 & \cdots & \omega^{2(N-1)} \\ 1 & \omega^3 & \omega^6 & \omega^9 & \cdots & \omega^{3(N-1)} \\ \vdots & \vdots & \vdots & \vdots & \ddots & \vdots \\ 1 & \omega^{N-1} & \omega^{2(N-1)} & \omega^{3(N-1)} & \cdots & \omega^{(N-1)(N-1)} \end{bmatrix} \qquad (2)$$

When mapping $\mathbf{W}$ on the memristor array, because every element of $\mathbf{W}$ is a complex number, $\mathbf{W}$ is decomposed into two matrixes of the same size with only real elements:

$$\mathbf{W} = \mathrm{Re}(\mathbf{W}) + \mathrm{Im}(\mathbf{W}) \qquad (3)$$

where $\mathrm{Re}(\mathbf{W})$ and $\mathrm{Im}(\mathbf{W})$ are the real part and imaginary part of $\mathbf{W}$, respectively. Similarly, $\mathbf{X}_k$ and $\mathbf{x}_n$ are also decomposed into real parts $\mathbf{X}_{k,\mathrm{real}}$, $\mathbf{x}_{n,\mathrm{real}}$ and imaginary parts $\mathbf{X}_{k,\mathrm{img}}$, $\mathbf{x}_{n,\mathrm{img}}$. Then, Eq. (1) can be rewritten as:

$$\begin{bmatrix} \mathbf{X}_{k,\mathrm{real}} \\ \mathbf{X}_{k,\mathrm{img}} \end{bmatrix} = \frac{1}{\sqrt{N}}\begin{bmatrix} \mathrm{Re}(\mathbf{W}) & -\mathrm{Im}(\mathbf{W}) \\ \mathrm{Im}(\mathbf{W}) & \mathrm{Re}(\mathbf{W}) \end{bmatrix} \cdot \begin{bmatrix} \mathbf{x}_{n,\mathrm{real}} \\ \mathbf{x}_{n,\mathrm{img}} \end{bmatrix} \qquad (4)$$

To perform the computation of Eq. (4) on MIR, the coefficient matrix is mapped on a memristor array as conductance and the input

$\mathbf{x}_n$ signal is represented as voltage pulses:

$$\begin{bmatrix} \mathbf{I}_{k,\text{real}} \\ \mathbf{I}_{k,\text{img}} \end{bmatrix} = \frac{1}{\sqrt{N}} \begin{bmatrix} \mathbf{G}_\text{R} & -\mathbf{G}_\text{I} \\ \mathbf{G}_\text{I} & \mathbf{G}_\text{R} \end{bmatrix} \cdot \begin{bmatrix} \mathbf{V}_{n,\text{real}} \\ \mathbf{V}_{n,\text{img}} \end{bmatrix} \tag{5}$$

In this manner, 1D DFT is completed. The computing paradigm is shown in Fig. 2e, f.

## Determination of the number of DFT point

According to our experimental results shown in Supplementary Fig. 4, with a greater number of DFT point, the computing results from MIR show higher consistency with software-computed results, indicating a better signal processing quality. Specifically, 8-point, 16-point, 32-point MIR DFT shows a correlation coefficient of 0.9908, 0.9980 and 0.9984 with software-computed results, respectively. In practice, the number of DFT point is still limited by the memristor array size. Here, using our customized test system with 16 K memristor array, we choose 64-point DFT for implementation in this work.

## Analysis of non-ideal device characteristics

To comprehensively evaluate the non-ideal device characteristics besides the mapping error, we run two simulations to analyze the impact of read noise and the stuck-at faults, respectively. As for the read noise, ten random scenarios with the read noise of 50-600 nA (standard deviation, STD) are simulated. As shown in Supplementary Fig. 2d and 2e, the present of read noise would eventually degrade the reconstructed image quality (PSNR) and segmentation accuracy (DICE). However, considering that the typical value of read noise STD is only 50-100 nA for our memristor devices, such degradation is unnoticeable because the simulation results show that the reconstructed image PSNR is still over 40 dB at the STD range of 50-100 nA (which means the degradation cannot be distinguished by human naked eyes). As for the stuck-at faults, seven random scenarios with the stuck-at fault range of 0.1%-2% are simulated. Considering the large randomness of stuck-at faults, we repeat every scenario for ten times and calculate the average value as the simulation results. As shown in Supplementary Fig. 6, a high yield of memristor arrays is critical and a large degradation of PSNR and DICE occurs at the stuck-at fault range of 0.1%-0.5%. After extensive efforts in the process development, the yield of our memristor arrays is as high as -99.99% (the percentage of stuck-at faults is -0.01%) and thus, the impact of stuck-at faults can be mostly ignored in this work. In addition, technically, we could also bypass the failed memristor cells when mapping the DFT matrices to mitigate the impact of stuck-at faults in practical applications.

## Analysis of interconnect resistance

The memristor array was fabricated with a standard 0.13μm Si CMOS process and the wire resistance between two adjacent memristors is estimated to be -0.1Ω, based on the PDK information provided by the foundry. It is concluded in our previous work[46], that the wire resistance does not have a serious impact on the computation accuracy for the 0.13μm technology node, and it starts to dominate the computation accuracy when the technology node is advanced to 14 nm or 28 nm. In such cases, techniques such as error balancing, bootstrapping and IR-drop adaptive compensation, can be used to mitigate the impact of wire interconnect resistance (e.g., refs. [47,48]).

## Implementation of 1D IDFT, 2D DFT and 2D IDFT

The expression of 1D IDFT is given as:

$$\mathbf{x}_n = \sqrt{N}\mathbf{W}^{-1}\cdot\mathbf{X}_k \tag{6}$$

where $\mathbf{W}^{-1}$ is the inverse matrix of $\mathbf{W}$. Because the DFT matrix is unitary, the inverse matrix, by definition, equals to the conjugate transpose matrix. Due to the symmetry characteristics, the inverse matrix of

DFT matrix equals the conjugate matrix $\mathbf{W}^*$:

$$\mathbf{x}_n = \sqrt{N}\mathbf{W}^{-1}\cdot\mathbf{X}_k = \sqrt{N}\mathbf{W}^*\cdot\mathbf{X}_k \tag{7}$$

Separating the real and imaginary parts, Eq. (7) can be rewritten as:

$$\begin{bmatrix} \mathbf{x}_{n,\text{real}} \\ \mathbf{x}_{n,\text{img}} \end{bmatrix} = \sqrt{N} \begin{bmatrix} \text{Re}(\mathbf{W}) & \text{Im}(\mathbf{W}) \\ -\text{Im}(\mathbf{W}) & \text{Re}(\mathbf{W}) \end{bmatrix} \cdot \begin{bmatrix} \mathbf{X}_{k,\text{real}} \\ \mathbf{X}_{k,\text{img}} \end{bmatrix} \tag{8}$$

When mapping on the memristor array, the computation can be performed as follows:

$$\begin{bmatrix} \mathbf{I}_{n,\text{real}} \\ \mathbf{I}_{n,\text{img}} \end{bmatrix} = \sqrt{N} \begin{bmatrix} \mathbf{G}_\text{R} & \mathbf{G}_\text{I} \\ -\mathbf{G}_\text{I} & \mathbf{G}_\text{R} \end{bmatrix} \cdot \begin{bmatrix} \mathbf{V}_{k,\text{real}} \\ \mathbf{V}_{k,\text{img}} \end{bmatrix} \tag{9}$$

Therefore, IDFT can then be implemented on MIR in a similar fashion as DFT, with only a minor difference on the sign of the imaginary part of DFT matrix.

In addition, 2D DFT and 2D IDFT for an $M \times N$ image $\mathbf{f}$ are mathematically defined as follows:

$$\mathbf{F}[k,l] = \frac{1}{MN} \sum_{m=0}^{M-1} \sum_{n=0}^{N-1} \mathbf{f}[m,n]e^{-j2\pi\left(\frac{k}{M}m + \frac{l}{N}n\right)} \tag{10}$$

$$\mathbf{f}[m,n] = \sum_{k=0}^{M-1} \sum_{l=0}^{N-1} \mathbf{F}[k,l]e^{j2\pi\left(\frac{k}{M}m + \frac{l}{N}n\right)} \tag{11}$$

where $k, m = 0,1,\cdots,M-1$ and $l, n = 0,1,\cdots,N-1$. $\mathbf{F}$ is an $M \times N$ 2D Fourier space signal. Such 2D DFT can be implemented by performing 2 steps of 1D DFT. As illustrated in Supplementary Fig. 5, the input image is first divided into column vectors and 1D DFT is implemented for each vector. The computing results of 1D DFT are composed into an intermediate matrix. After transpose, the intermediate matrix is again divided into column vectors and a second step of 1D DFT is performed, whose output represents the results of 2D DFT. 2D IDFT can be implemented in the same way as 2D DFT.

## Implementation of QAM and QM

As shown in Supplementary Fig. 1a, in the QAM strategy, a high mapping precision can be achieved because the mapping error is only introduced by the deliberately defined mapping margin in consideration of time overhead. Only the device whose conductance is within the mapping margin could successfully pass the mapping process, where a typical mapping margin is within 0.25 μS of the target value. By contrast, the QM strategy shown in Supplementary Fig. 1b often results in much lower mapping precision because the overall mapping (Supplementary Fig. 1e) error consists of two parts, the quantization error (Supplementary Fig. 1c) and the error due to pre-defined mapping margin (Supplementary Fig. 1d).

## MRI image reconstruction

The left atrium MRI dataset used in our work is one of the ten datasets in the Medical Segmentation Decathlon. The size of each MRI image is 320 × 320 and each dataset contains dozens of images. We randomly choose No.16 dataset to demonstrate the complete MRI reconstruction and segmentation. 19 other datasets (No. 3, 4, 5, 7, 9, 10, 11, 14, 16, 17, 18, 19, 20, 21, 22, 23, 24, 26, 29, 30) are used to further prove the consistent performance of MIR, as shown in Fig. 3i, j. Here, to simulate the MRI image reconstruction process, we first divide left atrium MRI image into several 64 × 64 patches for 64-point DFT computations on our memristor array. Then we perform lossless 2D DFT to transform the patches back into K-space. These K-space data are then considered as the raw MRI data and delivered to MIR to perform 2D IDFT. After 2D

IDFT, the transformed patches are stacked together to restore complete MRI images.

## CT image reconstruction

To simulate the actual CT image reconstruction process, we perform ideal Radon transform[49,50] to convert CT images into simulated projections and choose the reconstruction algorithm based on Fourier central slice theorem[51]. Technically, we first divide every CT image into several patches of $36 \times 36$. Then we perform the Radon transform from 180 angles in MATLAB, and the length of each projection vector is 64. For the $64 \times 180$ projection matrix, 64-point DFT is implemented on MIR. The transformed output vector of each projection vector is 64-point long and is then filled into a $64 \times 64$ 2D Fourier space. After the 64-point 2D IDFT implemented on MIR, the $64 \times 64$ 2D Fourier space is converted into a $64 \times 64$ spatial-domain image. At last, the $64 \times 64$ image is cropped into a $36 \times 36$ patch as a part of the reconstructed CT image. In addition, an overlap strategy is also used in CT reconstruction to obtain the best reconstruction quality. Here, we use a famous spleen CT image dataset from Memorial Sloan Kettering Cancer Center. Each dataset contains about 100 slices and the size of each slice is $512 \times 512$. The pixel value represents the Hounsfield Unit (HU) value. More details of CT scan criteria can be found in ref. [52].

## AI biomedical image segmentation

In this work, we implement nnU-Net on a GeForce GTX 1080 Ti GPU server with CUDA compilation tools 10.1. The nnU-Net's source code (which can be found on GitHub https://github.com/MIC-DKFZ/nnUNet) runs on Python 3.8.8 with PyTorch framework 1.7.1. To perform left atrium and spleen segmentations, we take the pretrained models for 3D semantic image segmentation with nnU-Net (can be found on Zenodo https://zenodo.org/record/3734294).

## Benchmarks of image quality and DICE score

We use two widely accepted metrics, PSNR and SNR, to evaluate the fidelity of reconstructed images. The definition of PSNR is related to mean squared error (MSE). Given an ideal $m \times n$ monochrome image $\mathbf{I}$ and the reconstructed noisy image $\mathbf{K}$, the mathematical expressions of MSE and PSNR (dB)[53] are:

$$\text{MSE} = \frac{1}{mn} \sum_{i=0}^{m-1} \sum_{j=0}^{n-1} [\mathbf{I}(i,j) - \mathbf{K}(i,j)]^2 \tag{12}$$

$$\text{PSNR} = 20 \log_{10} \left( \frac{\text{MAX}_{\mathbf{I}}}{\sqrt{\text{MSE}}} \right) \tag{13}$$

where $\text{MAX}_{\mathbf{I}}$ represents the maximum pixel value of image $\mathbf{I}$. The mathematical expression of SNR is:

$$\text{SNR} = 10 \log_{10} \left( \frac{\sum_{i=0}^{m-1} \sum_{j=0}^{n-1} \mathbf{I}(i,j)^2}{\sum_{i=0}^{m-1} \sum_{j=0}^{n-1} [\mathbf{I}(i,j) - \mathbf{K}(i,j)]^2} \right) \tag{14}$$

DICE score, ranging from 0 to 1, is widely used to measure whether two sets of data match well. Here, for the segmentation tasks, 1 corresponds to a perfect match between the segmented results and the ground truth, while 0 corresponds to no match. The expression of DICE score is given as follows[54]:

$$\text{DICE} = \frac{2|\mathbf{X} \cap \mathbf{Y}|}{|\mathbf{X}| + |\mathbf{Y}|} \tag{15}$$

where $\mathbf{X}$ and $\mathbf{Y}$ represent two different datasets (segmented results).

## Benchmark of energy efficiency

In the MRI image reconstruction task, an MRI raw data matrix with a size of $320 \times 320$ is used as the standard input data for estimating the energy efficiency. In total, 640 320-point IDFTs are carried out on MIR, consuming 47.9 μJ (the detailed energy consumption breakdown of each 320-point IDFT is shown in Supplementary Table 1a, which is obtained by XPEsim[55] using 65 nm technology). The transpose operation is reasonably neglected here because of its low computational cost. Thus, the overall number of required operations for one MRI image reconstruction is $640 \times 320 \times 320 \times 2 \times 4 = 524{,}288{,}000$, and the energy efficiency of MIR in MRI reconstruction task is 10.9 TOPS·W$^{-1}$ (i.e., 22,958 frames·W$^{-1}$). In comparison, the energy efficiency of representative Intel 12$^{th}$-Gen i9-12900 CPU and Nvidia Tesla V100 GPU is 9.5 GOPS·W$^{-1}$ (i.e., 19.3 frames·W$^{-1}$) and 100 GOPS·W$^{-1}$ (i.e., 205 frames·W$^{-1}$) and our MIR shows 1190× and 112× advantages, respectively.

In the CT image reconstruction task, a $768 \times 180$ projection matrix is used as the standard input raw data. First, 180 768-point DFTs are carried out on MIR. This step consumes a total of 57.0 μJ (as shown in Supplementary Table 1b, each 768-point DFT/IDFT consumes 316.8 nJ). Next, a 2D IDFT, which contains 1536 768-point IDFT, are computed on MIR, consuming 486.5 μJ. To sum up, the CT reconstruction process for one CT image consumes 543.5 μJ and requires $180 \times 768 \times 768 \times 2 \times 4 + 1536 \times 768 \times 768 \times 2 \times 4 = 8{,}097{,}103{,}872$ operations. As a result, the energy efficiency of MIR in CT reconstruction task is 14.9 TOPS·W$^{-1}$ (i.e., 2023 frames·W$^{-1}$), showing 1610× and 153× times advantages over Intel 12$^{th}$-Gen i9-12900 CPU (i.e., 1.25 frames·W$^{-1}$) and Nvidia Tesla V100 GPU (i.e., 13.2 frames·W$^{-1}$), respectively.

## Benchmark of the normalized image reconstruction speed

With highly efficient pipelines, each 1-bit-input VMM could take only 10 ns in our MIR. In the MRI reconstruction task, with 11-bits input data, the peak computing power of MIR is $320 \times 320 \times 4 \times 2 / 110$ ns = 7.3 TOPS. The die area of activated MIR core is 5.2 mm$^2$ (details can be found in Supplementary Table 1a) and then the computational density is estimated as 1.4 TOPS·mm$^{-2}$, which is equivalent to a normalized image reconstruction speed of 2962 fps·mm$^{-2}$. In the CT reconstruction task, the peak computing power is $768 \times 768 \times 4 \times 2 / 110$ ns = 41.9 TOPS. The die area of activated MIR core for CT takes an area of 13.8 mm$^2$ (Supplementary Table 1b). Hence the computational density is estimated as 3.1 TOPS·mm$^{-2}$ and the normalized image reconstruction speed is 414 fps·mm$^{-2}$.

Then we still choose Intel 12th-Gen i9-12900 CPU and Nvidia Tesla V100 GPU for comparison. As for Intel 12th-Gen i9-12900 CPU, it could provide a computing power of 0.6 TOPS and it takes an area of 200 mm$^2$. The computational density is estimated as 3 GOPS·mm$^{-2}$ (6.3 fps·mm$^{-2}$ for MRI; 0.4 fps·mm$^{-2}$ for CT). As for Nvidia Tesla V100 GPU, which could deliver a computing power of 31.4 TOPS, with a large die area of 815 mm$^2$, the computational density is estimated as 39 GOPS·mm$^{-2}$ (81 fps·mm$^{-2}$ for MRI images; 5.2 fps·mm$^{-2}$ for CT images). Therefore, in terms of the normalized image reconstruction speed, our MIR for MRI shows about 471× and 36× advantages over CPU and GPU, respectively. Also, MIR for CT shows about 1017× and 79× advantages over CPU and GPU, respectively.

## Data availability

The datasets that we used for benchmark are publicly available. The source data for Figs. 2–4 have been deposited and provided at GitHub https://github.com/Tsinghua-LEMON-Lab/Medical-image-reconstruction. Additional data supporting the findings of this study are available from the corresponding authors upon reasonable request.

## Code availability

The codes that support the findings of this study are available at GitHub https://github.com/Tsinghua-LEMON-Lab/Medical-image-reconstruction. Additional codes are available from the corresponding authors upon reasonable request.

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

## Acknowledgements
This work was in part supported by the STI 2030-Major Projects 2022ZD0210200 (J.T.), National Natural Science Foundation of China 61974081 (J.T.), 91964104 (J.T.), and 62025111 (H.W.), the XPLORER Prize (H.W.) and the Center of Nanofabrication, Tsinghua University.

## Author contributions
Z.L., H.Z. and J.T. conceived and designed the experiments. H.Z., Z.L., Y.Z., J.L. and Q.Q. contributed to the simulation. H.Z., Z.L., P.Y., Y.X. and Y.L. performed the experiments with the help from B.G., H.Q., and H.W.. Z.L., H.Z., and J.T. wrote the paper. All authors discussed and reviewed the manuscript.

## Competing interests
The authors declare no competing interests.
