## [Peer Review File · Nature Communications]

Reviewers' comments:

Reviewer #1 (Remarks to the Author):

It is recommended to publish the submitted manuscript, "Energy-efficient high-fidelity image reconstruction with memristor arrays for medical diagnosis" following minor revisions to address comments below. Overall, the manuscript is a good presentation of novel and significant work, with few weaknesses.

Strengths of the manuscript: Realization of the application of memristor computing-in-memory image processing by discrete Fourier transform analyzed using simulation in Ref. [33] is novel and important. Development of quasi-analog mapping and complex matrix transfer schemes to overcome precision and efficiency limitations inherent in the Ref. [33] approach is also novel. Demonstration on MRI and CT scan image reconstruction tasks is key evidence for providing proof-of-principle. This work is significant, as the use of FFT and DFT algorithms are ubiquitous in image processing and communications. Implementations of fast, low-power approaches on emerging memristor based computing-in-memory systems will find many additional applications, beyond the scope of the present work. The methodology followed to achieve these results is sound and does meet the standards of evidence expected in new device application development. The methodology section and supplemental materials are excellent, very detailed; sufficient for the reader to understand what has been achieved in detail, and sufficient for the expert to replicate the work.

Potential weaknesses of the manuscript, questions for clarification or revisions: Most of the conclusions and claims are well-supported with thorough evidence. There is one question however, regarding the claim that the low-power achievement enables portable medical imaging technology. For background to the question, consider that the portable MRI in Ref [5] states, "The subsystems of the scanner currently consume a total of ~800 W and can all be operated from a standard power outlet. This includes ~400 W for the RF transmit, 50 W for the gradient amplifiers (10 A into 2 Ω for each coil at a 5% duty cycle and 40% efficiency), 200 W for the console electronics and 75 W for the computer." In the section "Methods: Benchmark of energy efficiency", the manuscript has that "compared to Nvidia Tesla V100 GPU, that MIR performs a factor x112 improved in energy use". Wouldn't this mean that on a system like that in Ref. [5], the approximate power use would be 75 W / 112 ~ 0.7 W, a reduction of about 74 W for computing? If the answer is 'yes', then for an overall power of 800 W, a 74 W reduction is certainly helpful, but it's only a lightbulb's worth after all. I wouldn't think it accurate to describe the power reduction from the computing efficiency improvements as offering "...great potential in low-power ...portable medical imaging system for future medical scenarios" (p.3, lines 88-89). Isn't 800W power draw already rather low and quite portable, as Ref.[5] demonstrates? Is 725 W significantly more portable?

Similarly, the CereTom system from Ref. [11] is a 1300W portable head CT system that runs on a rechargeable battery. The brochure-standard computer for this system is based on Intel® Core™ 2 Duo Processor 3 GHz, so presumably, the computer power usage is about the same, 70-80 W. So, with a x153 reduction, as reported in the Methods section, the power usage might drop as low as 0.5 W, but again, this is not the major power draw for the system. Therefore, could the authors address this question: Is the power consumption from image processing really a significant barrier to portability in these systems?

If the authors can clarify this claim (that the energy efficiency improvement enables portability), this would strengthen the manuscript, but even if the claim is revised or withdrawn, this does not compromise the main novelty and significance of the results. The manuscript would still represent important work for which it is recommended to publish following appropriate revision.

Reviewer #2 (Remarks to the Author):

In this manuscript, Zhao et. al. attempted to demonstrate “Energy-efficient high-fidelity image reconstruction with memristor arrays for medical diagnosis”. However, if I read their earlier published IEDM paper (Ref. 38), then I don’t find any novelty or any new concepts in this current manuscript. A similar thing has been implemented, sometimes, they are either plotted on a different scale or projected differently. However, if one compares the concepts behind these two papers, both are very similar. Therefore, I do not recommend publishing this manuscript in a reputed journal like Nature Communications. Some of my detailed concerns are given below...

1. What is the novel idea in this manuscript? Idea-wise, it seems to be a repetition of reference 38. The only new contribution I could see was using the proposed CIM technique for MRI while reference 38 used this technique for CT.

2. Fig. 2(b) and 2(C) of this manuscript are same as of IEDM Figs. 4(b) and (6), respectively. Again, Fig. 2(e) (of this manuscript) and Fig. 10 (of IEDM) are of the same concepts. Fig. 4(b)-2D IDFT of this manuscript and Fig. 17(a) of IEDM are the same.

3. The paper describes fabrication of 1T-1R array, however Fig. 2(d) is just a memristive array. It's not clear, which bit cell are the authors using. I assume it is just a memristive cell without any transistors?

4. How will the PSNR and DICE scores be impacted if a Gaussian Noise (to model process variations) or when stuck at faults are introduced?

5. The authors suggest merging arrays in Fig. 2(f) to reduce ADC and other peripheral circuitry overhead. When the size of a memristive array increases, both the sneak paths and wire resistance also get increased. Unfortunately, the authors don't consider these. Without considering these non-idealities, the proposed system is too ideal.

6. The manuscript just depicts a new application (MRI) for matrix multiplication on memristive arrays without considering many major non-idealities.

Reviewer #3 (Remarks to the Author):

The manuscript presents simulation and experimental results using memristor arrays for DFT implementation to pursue image reconstruction of MRI and CT scans. The topic is interesting, and the figures are visually appealing. However, the novelty and the details of the work need to be highlighted. As stated right from the abstract, the authors suggest that the novelty of this paper is three-fold: proposing a quasi-analogue mapping (QAM) method, a generic complex matrix transfer (CMT) scheme and the actual experimental demonstration of the memristive image reconstructor on the two medical tasks.

However, the details supporting these novelty claims are unclear. The quasi-analogue mapping (QAM) scheme has already been proposed and experimentally demonstrated by the same group in IEDM 2021 (reference 38). The generic complex matrix transfer scheme refers to Fig. 2f. It is unclear how that figure is different from Fig. 13 in reference 38. The experimental demonstration is indeed of interest and seems to be of larger size than the one in reference 38, but more details need to be provided. The paper utilizes 8 arrays with 2,000 devices fabricated in a 0.13 μ m process and it includes characterization of a typical memristor device. However, it is unclear how many devices

from the 16K array were actually used for the experimental demonstration. The device-related figures either show behavior from one representative device in Fig. 1 or for representative noise from 256 devices in supplementary Fig. 2. The figure captions should emphasize what are simulation results and what are experimental results. A section on the device metrics across the 16k array impacting this application would be helpful. For example, is device-to-device variability affecting the performance in any way? What is programming scheme was used for the devices?

Overall, the paper has potential and addresses an important need in the medical field, but the novelty with respect to previously published work and the experimental methods should be detailed before being considered for Nature Communications.

Response Letter to Reviewers' Comments

We sincerely appreciate the valuable time the reviewers have spent reviewing our manuscript and providing insightful comments and suggestions to help further improve the quality of our work. We believe we have addressed all the reviewers' comments and now the paper is more rigorous in content and clearer in presentation. Our point-by-point responses to the reviewers' comments are as follows.

Reviewer #1

General Comments:

It is recommended to publish the submitted manuscript, "Energy-efficient high-fidelity image reconstruction with memristor arrays for medical diagnosis" following minor revisions to address comments below. Overall, the manuscript is a good presentation of novel and significant work, with few weaknesses.

Strengths of the manuscript: Realization of the application of memristor computing-in-memory image processing by discrete Fourier transform analyzed using simulation in Ref. [33] is novel and important. Development of quasi-analog mapping and complex matrix transfer schemes to overcome precision and efficiency limitations inherent in the Ref. [33] approach is also novel. Demonstration on MRI and CT scan image reconstruction tasks is key evidence for providing proof-of-principle. This work is significant, as the use of FFT and DFT algorithms are ubiquitous in image processing and communications. Implementations of fast, low-power approaches on emerging memristor based computing-in-memory systems will find many additional applications, beyond the scope of the present work. The methodology followed to achieve these results is sound and does meet the standards of evidence expected in new device application development. The methodology section and supplemental materials are excellent, very detailed; sufficient for the reader to understand what has been achieved in detail, and sufficient for the expert to replicate the work.

Response:

We sincerely thank the reviewer for the highly positive review on our work and recognizing the strengths and significance of the proposed MIR system for energy-efficient high-fidelity image reconstruction. We also appreciate your recommendation for publication of our work in Nature Communications. Our detailed responses to your technical comments are provided below.

Technical Comments:

Potential weaknesses of the manuscript, questions for clarification or revisions: Most of the conclusions and claims are well-supported with thorough evidence. There is one question however, regarding the claim that the low-power achievement enables portable medical imaging technology. For background to the question, consider that the portable MRI in Ref [5] states, "The subsystems of the scanner currently consume a total of ~800 W and can all be operated from a standard power outlet. This includes ~400 W for the RF transmit, 50 W for the gradient

amplifiers (10 A into 2Ω for each coil at a 5% duty cycle and 40% efficiency), 200 W for the console electronics and 75 W for the computer." In the section "Methods: Benchmark of energy efficiency", the manuscript has that "compared to Nvidia Tesla V100 GPU, that MIR performs a factor x112 improved in energy use". Wouldn't this mean that on a system like that in Ref. [5], the approximate power use would be $75 \text{ W} / 112 \sim 0.7 \text{ W}$, a reduction of about 74 W for computing? If the answer is 'yes', then for an overall power of 800 W, a 74 W reduction is certainly helpful, but it's only a lightbulb's worth after all. I wouldn't think it accurate to describe the power reduction from the computing efficiency improvements as offering "...great potential in low-power ...portable medical imaging system for future medical scenarios" (p.3, lines 88-89). Isn't 800W power draw already rather low and quite portable, as Ref.[5] demonstrates? Is 725 W significantly more portable?

Similarly, the CereTom system from Ref. [11] is a 1300W portable head CT system that runs on a rechargeable battery. The brochure-standard computer for this system is based on Intel® Core™ 2 Duo Processor 3 GHz, so presumably, the computer power usage is about the same, 70-80 W. So, with a x153 reduction, as reported in the Methods section, the power usage might drop as low as 0.5 W, but again, this is not the major power draw for the system. Therefore, could the authors address this question: Is the power consumption from image processing really a significant barrier to portability in these systems?

If the authors can clarify this claim (that the energy efficiency improvement enables portability), this would strengthen the manuscript, but even if the claim is revised or withdrawn, this does not compromise the main novelty and significance of the results. The manuscript would still represent important work for which it is recommended to publish following appropriate revision.

Response:

Thank you very much for your comment, raising an important question regarding the low power benefit of our MIR. Firstly, besides the power consumption as you mentioned, energy consumption should also be considered for portable medical imaging technology. Typical medical imaging systems, including MRI and CT, contain two independent steps: signal acquisition and image reconstruction (as described in our manuscript Page 2, Lines 37-44). These two steps usually take quite different working time scales and the power of each step doesn't exactly represent their actual energy consumptions. Take the CT imaging system as an example. As reported by the references (Boone, *Radiology*, 2006; Eklund, et al., *Med Image Anal*, 2013), a normal X-ray scanner would take only 0.08-0.30 seconds to rapidly acquire raw projection data, while even an advanced computer would take a much longer time to process those data, for example, 5-10 seconds reported by the reference (Yu, et al., *Journal of Signal Processing Systems*, 2019). Assuming the signal acquisition step consumes 1200W and the image reconstruction step consumes 70W, we could make a rough estimation that the signal acquisition step consumes $1200\text{W} \times 0.3\text{s} = 360\text{J}$ while the image reconstruction step consumes $70\text{W} \times 5\text{s} = 350\text{J}$. Thus, the image reconstruction step actually takes ~50% of the total energy consumption, instead of 5% of the power consumption as you referred in the reference (Carlson and Yonas, *Journal of Neuroimaging*,

2012). Hence, the energy overhead of medical image reconstruction is certainly nonnegligible and the optimization of it is of great significance.

Secondly, the rapid development of medical imaging system is imposing more challenges on image reconstruction. On the one hand, the consistent demand of portability puts a stricter limitation on the energy consumption of medical imaging systems (Cooley, et al., *Nat Biomed Eng*, 2021; Mazurek, et al., *Nat Commun*, 2021; Guallart-Naval, et al., *Scientific Reports*, 2022). For example, beneficial from the application of power-free permanent magnet, the power consumption of current medical imaging system has already been reduced to a few hundred watts (Guallart-Naval, et al., *Scientific Reports*, 2022; Wald, et al., *JMRI*, 2020) and future systems would demand for a further reduction on the energy consumption. On the other hand, to meet the demand for better image quality and higher signal acquisition speed, the number of MRI radio frequency coils and CT detectors has been increased dramatically (Eklund, et al., *Med Image Anal*, 2013; Corea, et al., *Nat Commun*, 2016; Vilela, et al., *ACS Nano*, 2018), resulting in an explosive growth of raw data to be processed. At the same time, more computationally intensive algorithms, such as iterative reconstruction algorithms, are also being implemented, imposing critical challenges for computing hardware (Geyer, et al., *Radiology*, 2015). Therefore, a more advanced image reconstructor with lower power and higher energy efficiency could be extremely helpful for developing future medical imaging systems.

Thirdly, researchers in the field of medical imaging have also been actively pursuing the optimization of image reconstruction process. For example, extensive studies (Yu, et al., *Journal of Signal Processing Systems*, 2019; Inam, et al., *Computers in Biology and Medicine*, 2020; Rymarczyk, et al., *Journal of Physics: Conference Series*, 2021) have adopted GPU or FPGA to implement image reconstruction algorithms to improve the computing speed and reduce the energy consumption. Prior works (Ravishankar, et al., *Proceedings of the IEEE*, 2020; Zhang and Dong, *Journal of the Operations Research Society of China*, 2020; Zhou, et al., *Proceedings of the IEEE*, 2021) also proposed new algorithms based on deep learning and data-adaptive methods to optimize the image reconstruction process.

Therefore, we believe that reducing the power consumption of image reconstruction with newly developed hardware like MIR is still of great significance and interest to enhance the portability of medical imaging systems. To illustrate this point more clearly, we have made the following revisions in the manuscript:

- 1) On page 2, lines 46-48, we supplement several literatures to support our point: “[21] Ravishankar S, Ye JC, Fessler JA. Image Reconstruction: From Sparsity to Data-Adaptive Methods and Machine Learning. *Proceedings of the IEEE* 108, 86-109 (2020). [22] Zhang H-M, Dong B. A Review on Deep Learning in Medical Image Reconstruction. *Journal of the Operations Research Society of China* 8, 311-340 (2020). [23] Zhou SK, et al. A Review of Deep Learning in Medical Imaging: Imaging Traits, Technology Trends, Case Studies with Progress Highlights, and Future Promises. *Proceedings of the IEEE* 109, 820-838 (2021)”.
- 2) On page 2, lines 46-52 the sentence “Amid the slowdown of Moore’s law scaling, ... with physically separated computing and memory units, limiting their energy efficiency.” has been

rewritten as “Besides, more sophisticated reconstruction algorithms based on iteration, deep learning and data-adaptive methods are also being implemented. Amid the slowdown of Moore’s law scaling, such computationally intensive tasks impose critical challenges for conventional computing hardware based on von Neumann architecture with physically separated computing and memory units, limiting their energy efficiency. Thus, the speed and energy consumption of the image reconstruction step has become a serious bottleneck for the development of portable medical imaging systems”.

Reviewer #2

General comments:

In this manuscript, Zhao et. al. attempted to demonstrate “Energy-efficient high-fidelity image reconstruction with memristor arrays for medical diagnosis”. However, if I read their earlier published IEDM paper (Ref. 38), then I don’t find any novelty or any new concepts in this current manuscript. A similar thing has been implemented, sometimes, they are either plotted on a different scale or projected differently. However, if one compares the concepts behind these two papers, both are very similar. Therefore, I do not recommend publishing this manuscript in a reputed journal like Nature Communications.

Response:

Thank you for your comments. We have carefully revised the manuscript following your comments. Our detailed point-by-point responses to your technical comments are provided below.

Comment #1:

What is the novel idea in this manuscript? Idea-wise, it seems to be a repetition of reference 38. The only new contribution I could see was using the proposed CIM technique for MRI while reference 38 used this technique for CT.

Response:

Thank you for your comment. Here, we shall clarify two things about the novelty issue:

Firstly, there is a four-page length limit for IEDM conference paper, so a lot of details are usually not presented. To encourage full disclosure of experimental results, IEDM explicitly states that “Publication in the digest in no way precludes later publication of a fuller account of the work in another journal” (<https://www.ieee-iedm.org/electronic-submission>). There are actually many examples of journal publication of extended version of IEDM papers.

Secondly, we must clarify that this work is not a simple repetition of our previous IEDM paper (reference 38), but rather contains many new experiments and important demonstrations. No experimental data are reused from our previous IEDM paper. Here we explain the key advances of our newly submitted work:

- 1) In this work, a more complexed and challenging **64-point DFT, rather than 26-point DFT** (reference 38), is implemented. In practice, 64-point DFT has a much wider range of applications than 26-point DFT. With 64-point DFT, signals of any length can be processed more efficiently with less butterfly computation, and meanwhile the spectral leakage and picket fence effect of DFT can also be largely mitigated. However, the implementation of 64-point DFT on memristor array with high accuracy is more challenging because it requires high-performance memristor device, large-size high-yield memristor array and more precise mapping strategy. Besides, the hardware overhead of peripheral circuits increases rapidly as the number of DFT grows, limiting the further increase of DFT scale as well as the system energy efficiency. To solve

the above challenges, we have made three major advancements in this work beyond reference 38 to implement 64-point memristive DFT. Firstly, thanks to the high yield and good device uniformity, the size of the memristor array (**16Kb**) to implement 64-point DFT is greatly improved by **more than 6 times**, compared to that utilized in the previous IEDM paper (only **2.64Kb**). With a larger array, we could perform the DFT algorithm more efficiently and accurately. Secondly, the number of RRAM conductance levels in this work (**25 levels**) is also **two times** more than that of IEDM paper (**11 levels**). Thus, the efficacy of quasi-analog mapping (QAM) is further validated here and the experimental results show that a more notable improvement with QAM was obtained in this work. This is also beneficial from the excellent analog switching characteristics of our memristors. Thirdly, to mitigate the increasing hardware overhead, we propose the **complex matrix transfer (CMT) strategy** in this work to improve the matrix transfer efficiency, which is completely novel and not reported previously. By assembling the four same-size DFT matrices into an integrated one, the real part or imaginary part of DFT results can be directly obtained in a single step and the number of peripheral circuits such as ADCs can be reduced by **at least one half**, saving both energy and area cost.

- 2) On the application side, in addition to CT, we demonstrated a more advanced and widely used imaging task, **MRI image reconstruction**. Although with a different signal acquisition method and image reconstruction algorithm, our MIR could still produce high-fidelity reconstructed images, which shows MIR has the potential to be a **general-purpose image processor**. More importantly, we further employ **nnU-Net** to segment the reconstructed medical images in order to better quantify their quality. The **high DICE score** confidently validates the reconstructed image quality and hence **the feasibility of practical applications for MIR**.

To better illustrate our point, we have made a table below to explain the difference between this work and our previous IEDM paper.

Comparison	IEDM paper	This work
Number of DFT points	26	64
Memristor array size	2.64K	16K
Number of memristor conductance level	11	25
DFT computing accuracy (correlation)	0.9990	0.9993
Computing scheme	Conventional	CMT
Demonstration tasks	CT	CT and MRI
Medical image segmentation	No	Yes, nnU-Net
Energy efficiency improvements over GPU	11.5×	153×
Image reconstruction speed improvements over GPU	Not benchmarked	79×

To illustrate this point more clearly, we also made the revision as follows:

- 1) On page 4, lines 141-147, we have added a statement: “**In fact, according to our experimental results shown in Supplementary Fig. 4, the implementation of a greater number of DFT point on**

MIR shows even higher consistency with software-computed results, indicating a better signal processing quality. However, as the number of DFT point grows, the required number of memristor conductance levels also increases rapidly (for example, 64-point DFT requires 25 conductance levels), and thus the efficacy of our quantization-error-free QAM strategy can be more notable to improve the mapping precision and computing accuracy”.

- 2) On page 3, lines 104-105, we have rewritten the sentence “To demonstrate the functionality of the MIR system, ...” as “**To demonstrate the functionality of the MIR system, we first implement a 64-point one-dimensional (1D) complex DFT on the memristor arrays with 16K memristors**”.
- 3) On page 4, lines 120-125, the sentence “As shown in **Fig. 2f**, ..., saving both energy and area cost” is rewritten as “**As shown in Fig. 2f, by assembling the four same-size DFT matrices into an integrated matrix, both the real and imaginary parts of DFT results can be directly obtained in a single step because the addition and subtraction operations can be realized inside the memristor array rather than by peripheral circuits, reducing the computing latency. In the meanwhile, the number of peripheral circuits such as ADCs and buffers can also be reduced by at least one half, saving both energy and area cost**”.

Comment #2:

Fig. 2(b) and 2(C) of this manuscript are same as of IEDM Figs. 4(b) and (6), respectively. Again, Fig. 2(e) (of this manuscript) and Fig. 10 (of IEDM) are of the same concepts. Fig. 4(b)-2D IDFT of this manuscript and Fig. 17(a) of IEDM are the same.

Response:

Thank you for your comment. We shall clarify that all the figures in this manuscript are different from those in IEDM. We have plotted all the figures with new experimental data and no data are reused from our previous IEDM paper. Their differences are elaborated as follows:

- 1) **Fig. 2(b) and 2(C):** These two figures are not the same as IEDM Fig. 4(b) and Fig. 6. First, Fig. 2(b) is a new TEM image of our memristor array with higher resolution to clearly illustrate the device structure with TiN/TaO_x/HfO₂/TiN material stack. Second, compared with Fig. 6 of IEDM, Fig. 2(c) of this manuscript plots the remeasured analog switching characteristics on a new memristor device, exhibiting a slightly larger switching window and improved symmetry.
- 2) **Fig. 2(e):** Fig. 2(e) of this manuscript and Fig. 10 of IEDM are schematics (not experimental data) plotted in slightly different styles to illustrate the same concept. In particular, Fig. 2(e) of this manuscript shows additional details on the peripheral circuits including signal buffers, ADCs, adder and subtractor, in order to make a clear comparison between the conventional DFT implementation on memristor arrays and the novel CMT scheme proposed in this manuscript.
- 3) **Fig. 4(b) - 2D IDFT:** Fig. 4(b) – 2D IDFT is an ideal CT slice that is used only to illustrate the CT image reconstruction process, while Fig. 17(a) of IEDM is the experimental results of the reconstructed CT image with memristor arrays. So they are different.

To make our point more clearly, we have also made the following revisions:

- 1) On page 18, lines 572-574, in the caption of Fig. 2(e), the sentence “Conventional DFT

implementation on memristor arrays” is rewritten as “Conventional DFT implementation on memristor arrays with extra copies of peripheral circuits to implement the addition and subtraction operations as well as analogue-to-digital conversion (ADC)”.

- 2) On page 21, lines 604-607, in the caption of Fig. 4(b), the sentence “Sub-figures, from left to right, ... and reconstructed image” is rewritten as “Sub-figures, from left to right, are actual human organ (the pink disc represents the section where the CT slice is acquired), projections from CT X-ray scanner, 2D Fourier space signal (intermediate results) and reconstructed CT image”.

Comment #3:

The paper describes fabrication of 1T-1R array, however Fig. 2(d) is just a memristive array. It’s not clear, which bit cell are the authors using. I assume it is just a memristive cell without any transistors?

Response:

Thank you for your comment. We are actually using 1T1R memristor array rather than passive array without transistors. The memristor array was fabricated with a standard 0.13 μm Si CMOS process and every unit cell contains one memristor and one Si transistor. Fig. 2(d) was a simplified schematic of cross-point array to illustrate the implementation of discrete Fourier transform on the memristor array and thus the transistors were omitted here. To avoid any possible misunderstanding, we have now replotted Fig. 2(d) as **Figure R1** and revised its caption as follows.

Figure R1. Implementation of discrete Fourier transform (DFT) on memristor array. DFT matrix is initially mapped onto a memristor array as the conductance. Then time-domain signals are fed into the bit lines (BLs) of memristor array as voltage pulses and frequency-domain signals are calculated as the output currents from the source lines (SLs). The word lines (WLs) are connected to the gate of transistors to select memristor cells.

Comment #4:

How will the PSNR and DICE scores be impacted if a Gaussian Noise (to model process variations) or when stuck at faults are introduced?

Response:

Thank you for your insightful comment. In fact, the Gaussian Noise introduced by process variation

could be mostly compensated by the precise QAM mapping process developed in this work and thus has little impact on the subsequent computations. Instead, we found that the Gaussian Noise in the read process (which is usually called, the read noise) could influence the computation results. Therefore, following your comment, we have run two new simulations to analyze the impact of read noise and the stuck-at faults, respectively, on the MRI image reconstruction task. The simulation results (**Figure R2 and Figure R3**) are as follows and have been added to the Supplementary Information as **Supplementary Figure 2d, 2e** and **Supplementary Figure 6**. As we can see, the presence of read noise would eventually degrade the reconstructed image quality (PSNR) and segmentation accuracy (DICE). However, considering that the typical value of standard deviation (STD) of read noise is only 50-100nA for our memristor devices, such degradation is unnoticeable because the simulation results show that the reconstructed image PSNR is still over 40dB at the STD range of 50-100nA (which means the degradation cannot be distinguished by human naked eyes). As for the stuck-at faults, a high yield of memristor arrays is critical and a serious degradation of PSNR and DICE occurs at the range of 0.1%-0.5%. After extensive efforts in the process development, the yield of our memristor arrays is ~99.99% (the percentage of stuck-at faults is ~0.01%) and thus, the impact of stuck-at faults can be mostly ignored in this work. In addition, technically, we could also bypass the failed memristor cells when mapping the DFT matrices to mitigate the impact of stuck-at faults in practical applications.

To clarify this point, we have revised the manuscript as follows:

- 1) On page 9, lines 286-302, we have added a section in **Methods**: “**Analysis of non-ideal device characteristics**. To comprehensively evaluate the non-ideal device characteristics besides the mapping error, we run two simulations to analyze the impact of read noise and the stuck-at faults, respectively. As for the read noise, ten random scenarios with the read noise of 50-600nA (standard deviation, STD) are simulated. As shown in **Supplementary Fig. 2d and 2e**, the present of read noise would eventually degrade the reconstructed image quality (PSNR) and segmentation accuracy (DICE). However, considering that the typical value of read noise STD is only 50-100nA for our memristor devices, such degradation is unnoticeable because the simulation results show that the reconstructed image PSNR is still over 40dB at the STD range of 50-100nA (which means the degradation cannot be distinguished by human naked eyes). As for the stuck-at faults, seven random scenarios with the stuck-at fault range of 0.1%~2% are simulated. Considering the large randomness of stuck-at faults, we repeat every scenario for ten times and calculate the average value as the simulation results. As shown in **Supplementary Fig. 6**, a high yield of memristor arrays is critical and a large degradation of PSNR and DICE occurs at the stuck-at fault range of 0.1%-0.5%. After extensive efforts in the process development, the yield of our memristor arrays is as high as ~99.99% (the percentage of stuck-at faults is ~0.01%) and thus, the impact of stuck-at faults can be mostly ignored in this work. In addition, technically, we could also bypass the failed memristor cells when mapping the DFT matrices to mitigate the impact of stuck-at faults in practical applications”.
- 2) On page 5, lines 170-173, the sentence “Here, in our work, ... reconstructed images by MIR” is revised as “Here, in this work, the non-ideal device characteristics of memristors and arrays, such as read noise (as shown in **Supplementary Fig. 2**), stuck-at faults (as shown in **Supplementary Fig. 6**), mapping error and interconnect resistance, could degrade the quality of reconstructed

images by MIR”.

- 3) On page 3, lines 39-42 of Supplementary Information, we have added a statement in the caption of **Supplementary Figure 2**: “Simulation results (**d**, PSNR; **e**, DICE) of different levels of read noise. It is suggested that the read noise would not seriously degrade the image quality in this work because the typical value of the standard deviation (STD) of read noise is 50-100nA for our memristor devices”.
- 4) On page 7, lines 70-75 of Supplementary Information, we have added a paragraph in the caption of **Supplementary Figure 6**: “**Supplementary Figure 6| Impact of memristor stuck-at faults.** Simulation results (**a**, PSNR; **b**, DICE) with different percentages of stuck-at faults, indicating the importance of achieving a high yield. A large degradation of PSNR and DICE occurs at the range of 0.1%-0.5%. However, after extensive efforts in the process development, the typical yield of our memristor arrays is ~99.99% (the percentage of stuck-at faults is ~0.01%). Hence, the impact of stuck-at faults can be mostly ignored in this work”.

Figure R2. Impact of memristor read noise. Simulation results (**d**, PSNR; **e**, DICE) of different levels of read noise. It is suggested that the read noise would not seriously degrade the image quality in this work because the typical value of the standard deviation (STD) of read noise is 50-100nA for our memristor devices.

Figure R3. Impact of memristor stuck-at faults. Simulation results (**a**, PSNR; **b**, DICE) with different percentages of stuck-at faults, indicating the importance of achieving a high yield. A large

degradation of PSNR and DICE occurs at the range of 0.1%-0.5%. However, after extensive efforts in the process development, the typical yield of our memristor arrays is ~99.99% (the percentage of stuck-at faults is ~0.01%). Hence, the impact of stuck-at faults can be mostly ignored in this work.

Comment #5:

The authors suggest merging arrays in Fig. 2(f) to reduce ADC and other peripheral circuitry overhead. When the size of a memristive array increases, both the sneak paths and wire resistance also get increased. Unfortunately, the authors don't consider these. Without considering these non-idealities, the proposed system is too ideal.

Response:

Thank you for your comment. As explained in the response to your Comment #3, we are using 1T1R memristor arrays in this work, so the sneak path problem is avoided. We can give a detailed explanation as follows. Firstly, during the SET or RESET process, we can accurately choose every single memristor without any sneak paths. In the memristor array architecture shown in **Figure R4**, if we want to program the conductance of the selected memristor in the figure, we first set the voltage of WL₁ as 'high' to turn on the NMOS transistor in the same row (red) and set the voltage of WL₂, WL₃ and WL₄ as 'low' to turn off the other transistors (black). The memristors in the other rows wouldn't introduce a sneak path because they are all "turned off" by the transistors connected in series with them. In addition to the selected memristor, the other memristors in the first row don't introduce sneak paths either, because the bit lines (BLs) BL₂, BL₃ and BL₄ are all left open or grounded and no current could flow through them. Consequently, during the SET or RESET process, no sneak path is introduced. Secondly, during the computation process, although all the transistors are turned on, there is no sneak path problem either. This is because all the BLs and SLs are clamped to a fixed voltage and there is no floating node in the array. Hence, the current of SL strictly follows Kirchhoff's current law and Ohm's law and thus, the array has no sneak paths during the computation process. To conclude, the 1T1R memristor array does not have the sneak path problem. More details about the sneak path could be found in references (Kannan, et al., *IEEE Transactions on Nanotechnology*, 2013; Zidan, et al., *Microelectronics Journal*, 2013; Cassuto, et al., *2013 IEEE International Symposium on Information Theory*, 2013).

Figure R4. Illustration of 1T1R memristor array architecture. Such memristor array does not have the sneak path problem.

As to the wire interconnect resistance, detailed simulation and analysis could be found in our previous

work (Liao, et al., *IEEE Transactions on CAD (TCAD)*, 2021). In this work, the memristor array was fabricated with a standard 0.13 μm Si CMOS process and the wire resistance between two adjacent memristors is estimated to be $\sim 0.1\Omega$, based on the PDK information provided by the foundry. It is concluded in our previous work (Liao, et al., *IEEE Transactions on CAD (TCAD)*, 2021), that such wire resistance does not have a serious impact on the computation accuracy for the 0.13 μm technology node, and it starts to dominate the computation accuracy when the technology node is advanced to 14nm or 28nm. In such cases, techniques such as error balancing, bootstrapping and IR-drop adaptive compensation, can be used to mitigate the impact of wire interconnect resistance (e.g., Mahmoodi, et al., *IEEE Transactions on Nanotechnology*, 2020; Liu, et al., *2014 IEEE/ACM International Conference on Computer-Aided Design (ICCAD)*, 2014).

To clarify this point, we have revised the manuscript as follows:

- 1) On page 9, lines 304-311, we have added a section in **Methods**: “**Analysis of interconnect resistance**. The memristor array was fabricated with a standard 0.13 μm Si CMOS process and the wire resistance between two adjacent memristors is estimated to be $\sim 0.1\Omega$, based on the PDK information provided by the foundry. It is concluded in our previous work, that the wire resistance does not have a serious impact on the computation accuracy for the 0.13 μm technology node, and it starts to dominate the computation accuracy when the technology node is advanced to 14nm or 28nm. In such cases, techniques such as error balancing, bootstrapping and IR-drop adaptive compensation, can be used to mitigate the impact of wire interconnect resistance (e.g., ref. ^{47, 48})”.
- 2) On page 15, lines 531-536, we have added three related references: “[46] Liao Y, *et al.* Diagonal Matrix Regression Layer: Training Neural Networks on Resistive Crossbars with Interconnect Resistance Effect. *IEEE Transactions on CAD (TCAD)* **40**, 1662-1671 (2021). [47] Mahmoodi MR, Vincent AF, Nili H, Strukov DB. Intrinsic Bounds for Computing Precision in Memristor-Based Vector-by-Matrix Multipliers. *IEEE Transactions on Nanotechnology* **19**, 429-435 (2020). [48] Liu B, *et al.* Reduction and IR-drop compensations techniques for reliable neuromorphic computing systems. In: *2014 IEEE/ACM International Conference on Computer-Aided Design (ICCAD)* (2014)”.

Comment #6:

The manuscript just depicts a new application (MRI) for matrix multiplication on memristive arrays without considering many major non-idealities.

Response:

Thank you for your comment. Again, we must argue that this manuscript contains many new experiments and important demonstrations, rather than just depicting a new application (MRI). The novelty of our work has been clarified in details in the response to your **Comments #1 and #2**. The supplementary analysis on non-idealities has been carried out in response to your **Comments #4 and #5**.

Reviewer #3

General comments:

The manuscript presents simulation and experimental results using memristor arrays for DFT implementation to pursue image reconstruction of MRI and CT scans. The topic is interesting, and the figures are visually appealing. However, the novelty and the details of the work need to be highlighted. As stated right from the abstract, the authors suggest that the novelty of this paper is three-fold: proposing a quasi-analogue mapping (QAM) method, a generic complex matrix transfer (CMT) scheme and the actual experimental demonstration of the memristive image reconstructor on the two medical tasks.

...

Overall, the paper has potential and addresses an important need in the medical field, but the novelty with respect to previously published work and the experimental methods should be detailed before being considered for Nature Communications.

Response:

We sincerely thank the reviewer for recognizing our work as “interesting and appealing”. We have carefully revised the manuscript following your insightful comments. Our detailed responses to your technical comments are provided below.

Comment #1:

However, the novelty and the details of the work need to be highlighted.

However, the details supporting these novelty claims are unclear. The quasi-analogue mapping (QAM) scheme has already been proposed and experimentally demonstrated by the same group in IEDM 2021 (reference 38).

Response:

Thank you for your comment. Here, we shall clarify two things about the novelty issue:

Firstly, there is a four-page length limit for IEDM conference paper, so a lot of details are usually not presented. To encourage full disclosure of experimental results, IEDM explicitly states that “Publication in the digest in no way precludes later publication of a fuller account of the work in another journal” (<https://www.ieee-iedm.org/electronic-submission>). There are actually many examples of journal publication of extended version of IEDM papers.

Secondly, we must clarify that this work is not a simple repetition of our previous IEDM paper (reference 38), but rather contains many new experiments and important demonstrations. No experimental data are reused from our previous IEDM paper. Here we explain the key advances of our newly submitted work:

- 1) In this work, a more complexed and challenging **64-point DFT, rather than 26-point DFT** (reference 38), is implemented. In practice, 64-point DFT has a much wider range of applications than 26-point DFT. With 64-point DFT, signals of any length can be processed more efficiently with less butterfly computation, and meanwhile the spectral leakage and picket fence effect of DFT can also be largely mitigated. However, the implementation of 64-point DFT on memristor array with high accuracy is more challenging because it requires high-performance memristor device, large-size high-yield memristor array and more precise mapping strategy. Besides, the hardware overhead of peripheral circuits increases rapidly as the number of DFT grows, limiting the further increase of DFT scale as well as the system energy efficiency. To solve the above challenges, we have made three major advancements in this work beyond reference 38 to implement 64-point memristive DFT. Firstly, thanks to the high yield and good device uniformity, the size of the memristor array (**16Kb**) to implement 64-point DFT is greatly improved by **more than 6 times**, compared to that utilized in the previous IEDM paper (only **2.64Kb**). With a larger array, we could perform the DFT algorithm more efficiently and accurately. Secondly, the number of RRAM conductance levels in this work (**25 levels**) is also **two times** more than that of IEDM paper (**11 levels**). Thus, the efficacy of quasi-analog mapping (QAM) is further validated here and the experimental results show that a more notable improvement with QAM was obtained in this work. This is also beneficial from the excellent analog switching characteristics of our memristors. Thirdly, to mitigate the increasing hardware overhead, we propose the **complex matrix transfer (CMT) strategy** in this work to improve the matrix transfer efficiency, which is completely novel and not reported previously. By assembling the four same-size DFT matrices into an integrated one, the real part or imaginary part of DFT results can be directly obtained in a single step and the number of peripheral circuits such as ADCs can be reduced by **at least one half**, saving both energy and area cost.
- 2) On the application side, in addition to CT, we demonstrated a more advanced and widely used imaging task, **MRI image reconstruction**. Although with a different signal acquisition method and image reconstruction algorithm, our MIR could still produce high-fidelity reconstructed images, which shows MIR has the potential to be a **general-purpose image processor**. More importantly, we further employ **nnU-Net** to segment the reconstructed medical images in order to better quantify their quality. The **high DICE score** confidently validates the reconstructed image quality and hence **the feasibility of practical applications for MIR**.

To better illustrate our point, we have made a table below to explain the difference between this work and our previous IEDM paper.

Comparison	IEDM paper	This work
Number of DFT points	26	64
Memristor array size	2.64K	16K
Number of memristor conductance level	11	25
DFT computing accuracy (correlation)	0.9990	0.9993
Computing scheme	Conventional	CMT
Demonstration tasks	CT	CT and MRI

Medical image segmentation	No	Yes, nnU-Net
Energy efficiency improvements over GPU	11.5×	153×
Image reconstruction speed improvements over GPU	Not benchmarked	79×

To illustrate this point more clearly, we also made the revision as follows:

- 1) On page 4, lines 141-147, we have added a statement: “**In fact, according to our experimental results shown in Supplementary Fig. 4, the implementation of a greater number of DFT point on MIR shows even higher consistency with software-computed results, indicating a better signal processing quality. However, as the number of DFT point grows, the required number of memristor conductance levels also increases rapidly (for example, 64-point DFT requires 25 conductance levels), and thus the efficacy of our quantization-error-free QAM strategy can be more notable to improve the mapping precision and computing accuracy**”.
- 2) On page 3, lines 104-105, we have rewritten the sentence “To demonstrate the functionality of the MIR system, ...” as “**To demonstrate the functionality of the MIR system, we first implement a 64-point one-dimensional (1D) complex DFT on the memristor arrays with 16K memristors**”.
- 3) On page 4, lines 120-125, the sentence “As shown in Fig. 2f, ..., saving both energy and area cost” is rewritten as “**As shown in Fig. 2f, by assembling the four same-size DFT matrices into an integrated matrix, both the real and imaginary parts of DFT results can be directly obtained in a single step because the addition and subtraction operations can be realized inside the memristor array rather than by peripheral circuits, reducing the computing latency. In the meanwhile, the number of peripheral circuits such as ADCs and buffers can also be reduced by at least one half, saving both energy and area cost**”.

Comment #2:

The generic complex matrix transfer scheme refers to Fig. 2f. It is unclear how that figure is different from Fig. 13 in reference 38.

Response:

Thank you for your comment. Fig. 13 in reference 38 illustrates the conventional DFT implementation on memristor arrays where the four DFT matrices are separated. Besides memristor arrays, such implementation would also need extra copies of peripheral circuits to implement the addition and subtraction operations as well as analogue-to-digital conversion (ADC), resulting in additional hardware overhead and computing latency, as shown in Fig. 2e. This is the key difference between these two figures. To circumvent this problem, a generic complex matrix transfer (CMT) scheme is proposed in this work, as shown in Fig. 2f. By assembling the four same-size DFT matrices into an integrated matrix, both the real and imaginary parts of DFT results can be directly obtained in a single step because the addition and subtraction operations can be realized inside the memristor array rather than by peripheral circuits. Specifically, the real part of DFT results $\text{Re}(y)$ equals to $\text{Re}(x)G_{\text{real}} + \text{Im}(x)(-G_{\text{img}})$ and the imaginary part $\text{Im}(y)$ equals to $\text{Re}(x)G_{\text{img}} + \text{Im}(x)G_{\text{real}}$. In fact, all these operations can be

realized at the same time by the assembled memristor array, thus greatly reducing the computing latency. In addition, the number of other peripheral circuits including buffers and ADCs can be reduced by at least one half (for example, 4 buffers and 4 ADCs are needed in the conventional implementation while only 2 buffers and 2 ADCs are required in the CMT scheme), saving both energy and area cost.

To make this point more clearly, we have revised our manuscripts as follows:

- 1) On page 4, lines 120-125, the sentence “As shown in **Fig. 2f**, ..., saving both energy and area cost” is rewritten as “**As shown in Fig. 2f, by assembling the four same-size DFT matrices into an integrated matrix, both the real and imaginary parts of DFT results can be directly obtained in a single step because the addition and subtraction operations can be realized inside the memristor array rather than by peripheral circuits, reducing the computing latency. In the meanwhile, the number of peripheral circuits such as ADCs and buffers can also be reduced by at least one half, saving both energy and area cost**”.

Comment #3:

The experimental demonstration is indeed of interest and seems to be of larger size than the one in reference 38, but more details need to be provided.

Response:

Thank you for your comment. In this work, a more complexed and challenging **64-point DFT, rather than 26-point DFT** (reference 38), is implemented. In practice, 64-point DFT has a much wider range of applications than 26-point DFT. With 64-point DFT, signals of any length can be processed more efficiently with less butterfly computation, and meanwhile the spectral leakage and picket fence effect of DFT can also be largely mitigated. However, the implementation of 64-point DFT on memristor array with high accuracy is more challenging because it requires high-performance memristor device, large-size high-yield memristor array and more precise mapping strategy. Besides, the hardware overhead of peripheral circuits increases rapidly as the number of DFT grows, limiting the further increase of DFT scale as well as the system energy efficiency. To solve the above challenges, we have made three major advancements in this work beyond reference 38 to implement 64-point memristive DFT. Firstly, thanks to the high yield and good device uniformity, the size of the memristor array (**16Kb**) to implement 64-point DFT is greatly improved by **more than 6 times**, compared to that utilized in the previous IEDM paper (only **2.64Kb**). With a larger array, we could perform the DFT algorithm more efficiently and accurately. Secondly, the number of RRAM conductance levels in this work (**25 levels**) is also **two times** more than that of IEDM paper (**11 levels**). Thus, the efficacy of quasi-analog mapping (QAM) is further validated here and the experimental results show that a more notable improvement with QAM was obtained in this work. This is also beneficial from the excellent analog switching characteristics of our memristors. Thirdly, to mitigate the increasing hardware overhead, we propose the **complex matrix transfer (CMT) strategy** in this work to improve the matrix transfer efficiency, which is completely novel and not reported previously. By assembling the four same-size DFT matrices into an integrated one, the real part or imaginary part of DFT results can be directly obtained in a single step and the number of peripheral circuits such as ADCs can be reduced

by **at least one half**, saving both energy and area cost. More detailed descriptions about the novelties and advances of this work can be found in our response to your **Comment #1**.

Comment #4:

The paper utilizes 8 arrays with 2,000 devices fabricated in a 0.13um process and it includes characterization of a typical memristor device. However, it is unclear how many devices from the 16K array were actually used for the experimental demonstration.

Response:

Thank you for your comment. In this work, we have implemented the 64-point DFT (the size of DFT coefficient matrix is 64×64). Four memristors are utilized to represent each one coefficient because the coefficient matrix has both real and imaginary parts with positive or negative values. Therefore, the required size of memristor array is $64 \times 64 \times 4 = 16384 = 16K$ and hence all the 16K memristors in our MIR system are actually used for the experimental demonstration. To make this point more clearly, we revised our manuscript as follows:

- 1) On page 3, lines 104-105, we have rewritten the sentence “To demonstrate the functionality of the MIR system, ...” as “**To demonstrate the functionality of the MIR system, we first implement a 64-point one-dimensional (1D) complex DFT on the memristor arrays with 16K memristors**”.
- 2) On page 9, lines 294-302, we have added a section in **Methods**: “**As for the stuck-at faults, seven random scenarios with the stuck-at fault range of 0.1%~2% are simulated. Considering the large randomness of stuck-at faults, we repeat every scenario for ten times and calculate the average value as the simulation results. As shown in **Supplementary Fig. 6**, a high yield of memristor arrays is critical and a large degradation of PSNR and DICE occurs at the stuck-at fault range of 0.1%-0.5%. After extensive efforts in the process development, the yield of our memristor arrays is as high as ~99.99% (the percentage of stuck-at faults is ~0.01%) and thus, the impact of stuck-at faults can be mostly ignored in this work. In addition, technically, we could also bypass the failed memristor cells when mapping the DFT matrices to mitigate the impact of stuck-at faults in practical applications**”.

Comment #5:

The device-related figures either show behavior from one representative device in Fig. 1 or for representative noise from 256 devices in supplementary Fig. 2. The figure captions should emphasize what are simulation results and what are experimental results.

Response:

Thank you for your comment. In fact, all the data in the original manuscript are experimental results except for those labeled as “software” (which is the software computing results serving as a comparison to our MIR computing results). However, in this revision, we have added some simulation data following the reviewers’ comments. Therefore, we have carefully revised our figure captions following your comment to clarify what are simulation results and what are experimental results:

- 1) On page 18, lines 576-577, the sentence in the caption of **Fig. 2g** “Top, ... strategy” is rewritten as “**Top, experimental mapping results with quasi-analogue mapping (QAM) strategy**”.
- 2) On page 18, lines 580, we added a sentence in the caption of **Fig. 2h**: “**MSE, mean squared error**”.
- 3) On page 3, lines 34-36 of Supplementary Information, the sentence in the caption of **Supplementary Figure 2** “Illustration ... measured for each level” is rewritten as “**Illustration of read noise of several representative conductance levels (256 devices are experimentally measured for each level)**”.
- 4) On page 3, lines 39-42 of Supplementary Information, we have added a statement in the caption of **Supplementary Figure 2**: “**Simulation results (d, PSNR; e, DICE) of different levels of read noise. It is suggested that the read noise would not seriously degrade the image quality in this work because the typical value of the standard deviation (STD) of read noise is 50-100nA for our memristor devices**”.
- 5) On page 5, lines 58-59 of Supplementary Information, we have added a sentence in the caption of **Supplementary Figure 4**: “**The circle represents the average correlation value of each scenario, while the error bar indicates the maximum and minimum values**”.
- 6) On page 7, lines 70-75 of Supplementary Information, we have added a paragraph in the caption of **Supplementary Figure 6**: “**Supplementary Figure 6| Impact of memristor stuck-at faults. Simulation results (a, PSNR; b, DICE) with different percentages of stuck-at faults, indicating the importance of achieving a high yield. A large degradation of PSNR and DICE occurs at the range of 0.1%-0.5%. However, after extensive efforts in the process development, the typical yield of our memristor arrays is ~99.99% (the percentage of stuck-at faults is ~0.01%). Hence, the impact of stuck-at faults can be mostly ignored in this work**”.

Comment #6:

A section on the device metrics across the 16k array impacting this application would be helpful. For example, is device-to-device variability affecting the performance in any way? What is programming scheme was used for the devices?

Response:

Thank you for your comment. We have added analysis on the impact of device non-idealities. Firstly, the impact of device-to-device variability is usually reflected in the mapping error, read noise and stuck-at faults. Therefore, besides the previously analyzed mapping error, we have now run several new simulations to precisely estimate the impact of read noise and stuck-at faults. Secondly, in addition to device-to-device variability, we have also analyzed the influence of interconnect resistance. Thirdly, we have added more information about our write-verify programming scheme. The results (**Figure R4** and **Figure R5**, which have been added to the Supplementary Information as **Supplementary Figure 2d, 2e** and **Supplementary Figure 6**) as well as revisions are as follows.

- 1) On page 9, lines 286-302, we have added a section in **Methods**: “**Analysis of non-ideal device characteristics. To comprehensively evaluate the non-ideal device characteristics besides the mapping error, we run two simulations to analyze the impact of read noise and the stuck-at faults, respectively. As for the read noise, ten random scenarios with the read noise of 50-600nA (standard**

deviation, STD) are simulated. As shown in **Supplementary Fig. 2d and 2e**, the present of read noise would eventually degrade the reconstructed image quality (PSNR) and segmentation accuracy (DICE). However, considering that the typical value of read noise STD is only 50-100nA for our memristor devices, such degradation is unnoticeable because the simulation results show that the reconstructed image PSNR is still over 40dB at the STD range of 50-100nA (which means the degradation cannot be distinguished by human naked eyes). As for the stuck-at faults, seven random scenarios with the stuck-at fault range of 0.1%~2% are simulated. Considering the large randomness of stuck-at faults, we repeat every scenario for ten times and calculate the average value as the simulation results. As shown in **Supplementary Fig. 6**, a high yield of memristor arrays is critical and a large degradation of PSNR and DICE occurs at the stuck-at fault range of 0.1%-0.5%. After extensive efforts in the process development, the yield of our memristor arrays is as high as ~99.99% (the percentage of stuck-at faults is ~0.01%) and thus, the impact of stuck-at faults can be mostly ignored in this work. In addition, technically, we could also bypass the failed memristor cells when mapping the DFT matrices to mitigate the impact of stuck-at faults in practical applications”.

- 2) On page 5, lines 170-173, the sentence “Here, in our work, ... reconstructed images by MIR” is revised as “Here, in this work, the non-ideal device characteristics of memristors and arrays, such as read noise (as shown in **Supplementary Fig. 2**), stuck-at faults (as shown in **Supplementary Fig. 6**), mapping error and interconnect resistance, could degrade the quality of reconstructed images by MIR”.
- 3) On page 3, lines 39-42 of Supplementary Information, we have added a statement in the caption of **Supplementary Figure 2**: “Simulation results (**d**, PSNR; **e**, DICE) of different levels of read noise. It is suggested that the read noise would not seriously degrade the image quality in this work because the typical value of the standard deviation (STD) of read noise is 50-100nA for our memristor devices”.
- 4) On page 7, lines 70-75 of Supplementary Information, we have added a paragraph in the caption of **Supplementary Figure 6**: “**Supplementary Figure 6| Impact of memristor stuck-at faults.** Simulation results (**a**, PSNR; **b**, DICE) with different percentages of stuck-at faults, indicating the importance of achieving a high yield. A large degradation of PSNR and DICE occurs at the range of 0.1%-0.5%. However, after extensive efforts in the process development, the typical yield of our memristor arrays is ~99.99% (the percentage of stuck-at faults is ~0.01%). Hence, the impact of stuck-at faults can be mostly ignored in this work”.
- 5) On page 9, lines 304-311, we have added a section in **Methods**: “**Analysis of interconnect resistance.** The memristor array was fabricated with a standard 0.13 μ m Si CMOS process and the wire resistance between two adjacent memristors is estimated to be ~0.1 Ω , based on the PDK information provided by the foundry. It is concluded in our previous work, that the wire resistance does not have a serious impact on the computation accuracy for the 0.13 μ m technology node, and it starts to dominate the computation accuracy when the technology node is advanced to 14nm or 28nm. In such cases, techniques such as error balancing, bootstrapping and IR-drop adaptive compensation, can be used to mitigate the impact of wire interconnect resistance (e.g., ref. ^{47, 48})”.
- 6) On page 15, lines 531-536, we added several literatures to support our point: “[46] Liao Y, *et al.* Diagonal Matrix Regression Layer: Training Neural Networks on Resistive Crossbars with

Interconnect Resistance Effect. *IEEE Transactions on CAD (TCAD)* **40**, 1662-1671 (2021). [47] Mahmoodi MR, Vincent AF, Nili H, Strukov DB. Intrinsic Bounds for Computing Precision in Memristor-Based Vector-by-Matrix Multipliers. *IEEE Transactions on Nanotechnology* **19**, 429-435 (2020). [48] Liu B, *et al.* Reduction and IR-drop compensations techniques for reliable neuromorphic computing systems. In: *2014 IEEE/ACM International Conference on Computer-Aided Design (ICCAD)* (2014)".

- 7) On page 8, line 249, the title “**Fabrication of 1T1R memristor array**” is rewritten as “**Fabrication and programming of 1T1R memristor array**”. On lines 254-258, we added some information about our write-verify programming scheme: “As for the programming of 1T1R memristor array, we use the standard write-verify programming scheme. Here, multiple voltage pulses are applied to the 1T1R memristor cell to increase (decrease) the conductance, until the conductance is larger (smaller) or equal to the target values. This process is repeated until the memristor conductance to programmed within the error margin of the target value”.

Figure R4. Illustration of memristor read noise. Simulation results (d, PSNR; e, DICE) of different levels of read noise. It is suggested that the read noise would not seriously degrade the image quality in this work because the typical value of the standard deviation (STD) of read noise is 50-100nA for our memristor devices.

Figure R5. Impact of memristor stuck-at faults. Simulation results (**a**, PSNR; **b**, DICE) with different percentages of stuck-at faults, indicating the importance of achieving a high yield. A large degradation of PSNR and DICE occurs at the range of 0.1%-0.5%. However, after extensive efforts in the process development, the typical yield of our memristor arrays is ~99.99% (the percentage of stuck-at faults is ~0.01%). Hence, the impact of stuck-at faults can be mostly ignored in this work.

References

- Boone JM. Multidetector CT: Opportunities, Challenges, and Concerns Associated with Scanners with 64 or More Detector Rows. *Radiology* **241**, 334-337 (2006).
- Eklund A, Dufort P, Forsberg D, LaConte SM. Medical image processing on the GPU – Past, present and future. *Med Image Anal* **17**, 1073-1094 (2013).
- Yu X, Wang H, Feng W-c, Gong H, Cao G. GPU-Based Iterative Medical CT Image Reconstructions. *Journal of Signal Processing Systems* **91**, 321-338 (2019).
- Carlson AP, Yonas H. Portable Head Computed Tomography Scanner–Technology and Applications: Experience with 3421 Scans. *Journal of Neuroimaging* **22**, 408-415 (2012).
- Cooley CZ, *et al.* A portable scanner for magnetic resonance imaging of the brain. *Nat Biomed Eng* **5**, 229-239 (2021).
- Mazurek MH, *et al.* Portable, bedside, low-field magnetic resonance imaging for evaluation of intracerebral hemorrhage. *Nat Commun* **12**, 5119 (2021).
- Gualart-Naval T, *et al.* Portable magnetic resonance imaging of patients indoors, outdoors and at home. *Scientific Reports* **12**, 13147 (2022).
- Wald LL, McDaniel PC, Witzel T, Stockmann JP, Cooley CZ. Low-cost and portable MRI. *JMRI* **52**, 686-696 (2020).
- Corea JR, *et al.* Screen-printed flexible MRI receive coils. *Nat Commun* **7**, 10839 (2016).
- Vilela D, *et al.* Medical Imaging for the Tracking of Micromotors. *ACS Nano* **12**, 1220-1227 (2018).
- Geyer LL, *et al.* State of the Art: Iterative CT Reconstruction Techniques. *Radiology* **276**, 339-357 (2015).
- Inam O, Basit A, Qureshi M, Omer H. FPGA-based hardware accelerator for SENSE (a parallel MR image reconstruction method). *Computers in Biology and Medicine* **117**, 103598 (2020).
- Rymarczyk T, Kosior A, Tchórzewski P, Vejar A. Image reconstruction in electrical impedance tomography using a reconfigurable FPGA system. *Journal of Physics: Conference Series* **1782**, 012033 (2021).
- Ravishankar S, Ye JC, Fessler JA. Image Reconstruction: From Sparsity to Data-Adaptive Methods and Machine Learning. *Proceedings of the IEEE* **108**, 86-109 (2020).
- Zhang H-M, Dong B. A Review on Deep Learning in Medical Image Reconstruction. *Journal of the Operations Research Society of China* **8**, 311-340 (2020).
- Zhou SK, *et al.* A Review of Deep Learning in Medical Imaging: Imaging Traits, Technology Trends, Case Studies with Progress Highlights, and Future Promises. *Proceedings of the IEEE* **109**, 820-838 (2021).
- Kannan S, Rajendran J, Karri R, Sinanoglu O. Sneak-Path Testing of Crossbar-Based Nonvolatile Random Access Memories. *IEEE Transactions on Nanotechnology* **12**, 413-426 (2013).
- Zidan MA, Fahmy HAH, Hussain MM, Salama KN. Memristor-based memory: The sneak paths problem and solutions. *Microelectronics Journal* **44**, 176-183 (2013).
- Cassuto Y, Kvatinsky S, Yaakobi E. Sneak-path constraints in memristor crossbar arrays. In: *2013 IEEE International Symposium on Information Theory* (2013).
- Liao Y, *et al.* Diagonal Matrix Regression Layer: Training Neural Networks on Resistive Crossbars With Interconnect Resistance Effect. *IEEE Transactions on CAD (TCAD)* **40**, 1662-1671 (2021).
- Mahmoodi MR, Vincent AF, Nili H, Strukov DB. Intrinsic Bounds for Computing Precision in Memristor-Based Vector-by-Matrix Multipliers. *IEEE Transactions on Nanotechnology* **19**, 429-435 (2020).
- Liu B, *et al.* Reduction and IR-drop compensations techniques for reliable neuromorphic computing systems. In: *2014 IEEE/ACM International Conference on Computer-Aided Design (ICCAD)* (2014).

REVIEWERS' COMMENTS

Reviewer #1 (Remarks to the Author):

The authors have adequately responded to my questions and concerns. I thank the authors for a thorough response and the associated changes to the manuscript. I recommend the manuscript for publication.

Reviewer #2 (Remarks to the Author):

In the modified version of the manuscript, the authors tried to address all the concerns which were raised earlier and also made substantial changes. The quality of the manuscript has been improved and this version can now be accepted for publication.

Response Letter to Reviewers' Comments

Reviewer #1 (Remarks to the Author):

The authors have adequately responded to my questions and concerns. I thank the authors for a thorough response and the associated changes to the manuscript. I recommend the manuscript for publication.

Response:

We sincerely thank you for your recommendation and the valuable time you have spent reviewing our manuscript and providing insightful comments to help significantly improve the quality of our work. We are very glad to see that you are satisfied with our revision.

Reviewer #2 (Remarks to the Author):

In the modified version of the manuscript, the authors tried to address all the concerns which were raised earlier and also made substantial changes. The quality of the manuscript has been improved and this version can now be accepted for publication.

Response:

We sincerely thank you for the valuable time you have spent reviewing our manuscript and providing insightful comments to help significantly improve the quality of our work. We are very glad to see that you are satisfied with our revision.